# ImpScore: A Learnable Metric For Quantifying The Implicitness Level of Sentences

**Yuxin Wang**[†]   **Xiaomeng Zhu**[⋆]   **Weimin Lyu**[◇]   **Saeed Hassanpour**[‡]   **Soroush Vosoughi**[†]

[†]Department of Computer Science, Dartmouth College
[⋆]Department of Linguistics, Yale University
[◇]Department of Computer Science, Stony Brook University
[‡]Department of Biomedical Data Science, Dartmouth College

## Abstract

Handling implicit language is essential for natural language processing systems to achieve precise text understanding and facilitate natural interactions with users. Despite its importance, the absence of a metric for accurately measuring the implicitness of language significantly constrains the depth of analysis possible in evaluating models' comprehension capabilities. This paper addresses this gap by developing a scalar metric that quantifies the implicitness level of language without relying on external references. Drawing on principles from traditional linguistics, we define "implicitness" as the divergence between semantic meaning and pragmatic interpretation. To operationalize this definition, we introduce IMPSCORE, a reference-free metric formulated through an interpretable regression model. This model is trained using pairwise contrastive learning on a specially curated dataset consisting of (*implicit sentence*, *explicit sentence*) pairs. We validate IMPSCORE through a user study that compares its assessments with human evaluations on out-of-distribution data, demonstrating its accuracy and strong correlation with human judgments. Additionally, we apply IMPSCORE to hate speech detection datasets, illustrating its utility and highlighting significant limitations in current large language models' ability to understand highly implicit content. Our metric is publicly available at https://github.com/audreycs/ImpScore.

## 1 Instruction

Implicit expression of opinions or attitudes is pervasive in interpersonal communication, presenting unique challenges and opportunities for Natural Language Processing (NLP). As large language models (LLMs) have scaled in parameter size, their capabilities in understanding implicit information have improved significantly, impacting tasks like automatic content moderation and AI safety (Hartvigsen et al., 2022; Kim et al., 2022; Wen et al., 2023).

Despite these advancements, critical foundational limitations remain, particularly in the nuanced understanding and measurement of implicit language. Current benchmarks often employ subjective criteria for collecting implicit data, which complicates the assessment of data quality and undermines robust evaluation (Jahan & Oussalah, 2023). Furthermore, the prevalent binary labeling of data as either implicit or explicit does not align well with the complex ways humans use and perceive language, obscuring the true depth of LLMs' understanding capabilities (Anwar et al., 2024). Addressing these challenges requires a more refined, granular metric for implicitness that can evaluate language on a continuum rather than in binary terms.

This paper introduces a novel, reference-free metric—IMPSCORE —to quantitatively assess the implicitness of any English sentence. **We define "implicitness" as the divergence between the semantic meaning and the pragmatic interpretation of a sentence**, where we use "semantic" to describe the literal meaning of a sentence, which is a function of the literal meaning of the words and phrases that compose the sentence and its syntactic structure (Partee, 1984). We use "pragmatic" to describe the intended meaning that can be inferred from context and usage. This definition is inspired by foundational work in theoretical linguistics (Grice, 1975; Carston, 2009). For instance, the sentence *"I'm finding it a bit hard to concentrate"* typically communicates difficulty in concentration

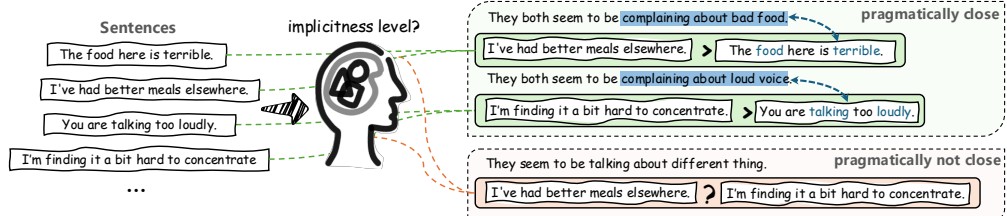

Figure 1: An illustration of how humans typically perceive sentences with different implicitness levels. Sentences directly expressing their pragmatics are generally more explicit (indicated by blue arrows), and it is easier to distinguish the implicitness levels between sentences with close pragmatics (expressing similar meaning).

but can pragmatically imply a complaint about the loudness of an addressee's speech, demonstrating a significant semantic-pragmatic divergence. In contrast, *"You are talking too loudly"* is more explicit, as the spoken content directly conveys the intended meaning. These are illustrated in Fig. 1.

Basing on this definition, we designed an interpretable regression model and trained a reference-free, learnable metric, **IMPSCORE**, to measure implicitness scores for English sentences. As shown in Fig. 2, for each input sentence, IMPSCORE first uses a text encoder to generate an embedding, then maps its semantic and pragmatic features into two independent low-dimensional spaces. The implicitness score is calculated by transforming these features into the same space and applying a distance metric to measure their divergence. Since our goal is to measure the implicitness of an arbitrary sentence, not just among sentences of the same intended meaning, IMPSCORE includes an additional indicator to measure pragmatic distance between sentences to aid the interpretation of their implicitness scores. This is illustrated in Fig. 1, where it is more difficult to compare implicitness level between sentences with dissimilar pragmatics (red box). IMPSCORE is trained via pairwise contrastive learning, where we collected and processed $112,580$ (*implicit sentence*, *explicit sentence*) pairs from various data sources, half of which are pragmatically close (*i.e.*, positive pairs) and half are pragmatically distinct (*i.e.*, negative pairs). IMPSCORE is tasked to generate higher implicitness scores for implicit sentences in each pair and smaller pragmatic distances for positive pairs.

IMPSCORE's training achieved a high accuracy rate, exceeding 95%, in both discerning levels of implicitness between sentences and differentiating degrees of pragmatic closeness among sentence pairs. To evaluate the effectiveness of our trained IMPSCORE on out-of-distribution (OOD) data, we implemented a user study designed to rank implicitness levels and measure pragmatic closeness. The results from this study indicated that IMPSCORE's predictions achieve reasonable accuracy and show a positive correlation with human judgments. Notably, IMPSCORE's performance consistently aligns with the Law of Large Numbers, demonstrating that as the number of observations increases, the average of the results becomes more accurate and stable across diverse datasets.

Further exploring the utility of IMPSCORE, we applied it in the context of hate speech detection across two main applications: (1) was utilized to evaluate popular datasets, analyzing their levels of implicitness and pragmatic diversity. Our quantitative analysis confirms the model's capability to effectively discern varying degrees of implicitness. (2) We assessed the performance of three advanced language models—GPT-4-Turbo, Llama-3.1-8B-Instruct, and OpenAI Moderation—against data exhibiting different levels of implicitness. The findings revealed a notable degradation in their performance as the implicitness level intensified, highlighting a critical bottleneck in these models' ability to comprehend highly implicit sentences, a challenge often obscured by their generally robust performances on standard hate speech detection benchmarks.

The lightweight nature of IMPSCORE, coupled with its capability for rapid inference, enables efficient and effective quantitative comparisons across extensive text corpora. These attributes position IMPSCORE as a preferable alternative to more costly LLM evaluators, which often raise concerns regarding their operational transparency, faithfulness, and computational efficiency.

**The key contributions of this paper are:**
- Introduction of a novel metric, IMPSCORE, which offers a refined interpretation of "implicitness," effectively quantifying this aspect of language in a learnable and operational manner.
- Construction of a comprehensive dataset comprising $112,580$ (*implicit sentence*, *explicit sentence*) pairs from diverse sources, tailored specifically to train IMPSCORE.
- Execution of a user study aimed at validating IMPSCORE's performance on OOD data, which confirmed the model's adeptness at accurately predicting sentence implicitness levels.

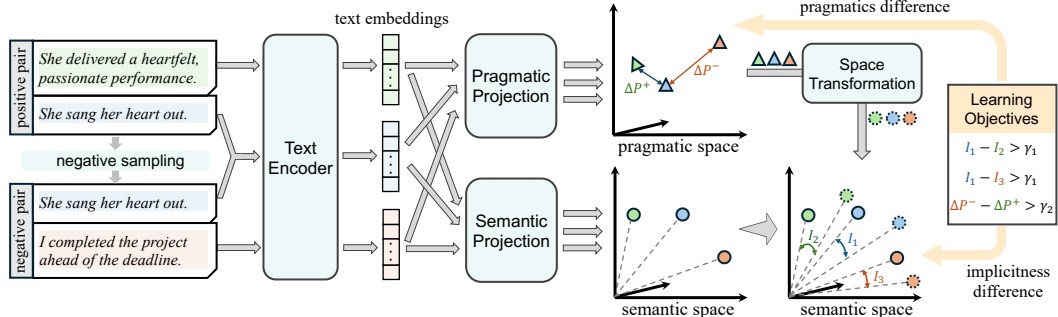

Figure 2: An overview of the training process of IMPSCORE. The sentence in blue is an implicit sentence, while the sentence in green and the sentence in red are explicit sentences in the positive and negative pairs, respectively. Colored $\triangle$ and $\bigcirc$ markers denote the feature embedding points of sentences in corresponding colors. $I_1$, $I_2$, and $I_3$ denote the implicitness scores of these sentences. $\Delta P^+$ and $\Delta P^-$ denote the pragmatic distances of the positive and negative pairs, respectively. $\gamma_1$ and $\gamma_2$ are model hyperparameters.

- Application of IMPSCORE in the realm of hate speech detection, revealing significant insights into the limitations of contemporary language models in processing highly implicit language.

## 2  RELATED WORK

**Automatic NLP Metrics on Text Features** Many rule-based and machine learning-based NLP metrics have been proposed to evaluate text quality and features, in either reference-based or reference-free manner. Representative works include those in machine translation (Papineni et al., 2002; Zhang et al., 2020), coherence and cohesion analysis (Barzilay & Lapata, 2008; Pradhan et al., 2011), and readability assessment (Flesch, 1948; Lee & Vajjala, 2022). Recently, there has been work delving into quantifying unique text characteristics, such as simplicity (Cripwell et al., 2023), politeness (Fu et al., 2020), and factual precision (Min et al., 2023). However, to the best of our knowledge, there is no existing automatic metric on evaluating implicitness of text. Directly applying other metrics is also ill-suited to address this problem.

**Linguistic Measures of Implicitness** Theoretical linguistics has long examined how exactly meaning is conveyed through phenomena such as presupposition (Beaver, 1997) and implicature (Grice, 1975), without providing a quantitative measure of the implicitness of sentences. Among the few relevant works, Vallauri & Masia (2014), proposed a system of quantification indexes to measure implicit causality bias, focusing on how specific verbs influence causal attributions by calculating bias scores for each verb. Garassino et al. (2022) qualitatively assessed participants' abilities to detect and interpret different types of implicit content. However, neither study addressed calculating implicitness scores for entire sentences. Additionally, their results heavily rely on human annotations, making them difficult to reproduce automatically.

## 3  FRAMEWORK OF IMPSCORE

Fig. 2 illustrates the overall training process of IMPSCORE, which employs pairwise contrastive learning and computes an implicitness score $I$ for each sentence. Each step is detailed below.

### 3.1  POSITIVE AND NEGATIVE PAIRS PREPARATION

The input to IMPSCORE consists of a **positive pair** and a corresponding **negative pair**. The positive pair contains an implicit sentence $s_1$ and a pragmatically close explicit sentence $s_2$, while the negative pair uses the same $s_1$ but replaces $s_2$ with a pragmatically distant sentence $s_3$. In this design, $s_1$ serves as an anchor, ensuring a pragmatically close-far relationship between the two pairs for IMPSCORE to learn. The methodology for constructing these pairs is detailed in §4.

### 3.2  PAIRWISE LEARNING OF IMPLICITNESS

After inputting the prepared positive pair $(s_1, s_2)$ and negative pair $(s_1, s_3)$ into IMPSCORE, IMPSCORE generates text embeddings $\mathbf{e}_1$, $\mathbf{e}_2$, and $\mathbf{e}_3 \in \mathbb{R}^d$ for $s_1$, $s_2$, and $s_3$ respectively using a Text Encoder $f_{\boldsymbol{\theta}}$. Here, $d$ is the embedding dimension and $\boldsymbol{\theta}$ denotes the parameters of Text Encoder:

$$\mathbf{e}_{\{1,2,3\}} = f_{\boldsymbol{\theta}}(s_{\{1,2,3\}}). \tag{1}$$

Then, IMPSCORE extracts the semantic features and pragmatic features from the text embeddings. We use two projection matrices to map text embeddings into a pragmatic and a semantic space independently — a Pragmatic Projection matrix $\mathbf{W}_p \in \mathbb{R}^{d \times l}$ and a Semantic Projection matrix $\mathbf{W}_s \in \mathbb{R}^{d \times l}$. $l$ is the dimension of semantic and pragmatic features which is much smaller than $d$. The pragmatic feature $\mathbf{h}^p$ and semantic feature $\mathbf{h}^s$ for $s_1$, $s_2$, and $s_3$ are derived via below multiplications:

$$\mathbf{h}^p_{\{1,2,3\}} = \mathbf{e}_{\{1,2,3\}}\mathbf{W}_p, \qquad \mathbf{h}^s_{\{1,2,3\}} = \mathbf{e}_{\{1,2,3\}}\mathbf{W}_s. \tag{2}$$

To measure the implicitness of each sentence, which is interpreted as the divergence between its pragmatics and semantics, we then put the two features in the same space. We simply transform the pragmatic features into the semantic space using a *Space Transformation* matrix $\mathbf{W}_t \in \mathbb{R}^{l \times l}$ (other transformation approaches are also explored in §5.3.):

$$\hat{\mathbf{h}}^s_{\{1,2,3\}} = \mathbf{h}^p_{\{1,2,3\}}\mathbf{W}_t, \tag{3}$$

where $\hat{\mathbf{h}}^s_1, \hat{\mathbf{h}}^s_2, \hat{\mathbf{h}}^s_3 \in \mathbb{R}^l$ are transformed pragmatic features of $s_1$, $s_2$, and $s_3$ in semantic space. Then, the level of implicitness $I$ of each sentence is calculated as the Cosine distance of the transformed semantic features and original semantic features. Employing Cosine distance as the metric ensures that the implicitness score remains within a standardized range of $[0, 2]$, regardless of changes in the model parameters.

$$I_{\{1,2,3\}} = 1 - \texttt{CosineSim}(\mathbf{h}^s_{\{1,2,3\}}, \hat{\mathbf{h}}^s_{\{1,2,3\}}). \tag{4}$$

The objective for this pairwise learning of implicitness is to maximize the implicitness level of $s_1$, and minimize the levels of $s_2$ and $s_3$. We define the implicitness loss $\mathcal{L}_{imp}$ as below, which adopts Pairwise Margin-Based Ranking Loss:

$$\begin{aligned} \mathcal{L}_{imp}(s_1, s_2) &= \texttt{max}(0, \gamma_1 - (I_1 - I_2)), \\ \mathcal{L}_{imp}(s_1, s_3) &= \texttt{max}(0, \gamma_1 - (I_1 - I_3)), \end{aligned} \tag{5}$$

where $\gamma_1$ is a hyperparameter with range $[0, 2)$. By minimizing the two losses, the model is encouraged to compute implicitness level for $s_1$ at least $\gamma_1$ higher than that of $s_2$ and $s_3$.

### 3.3 PAIRWISE LEARNING OF PRAGMATICS

As outlined in the introduction, we also want to learn an indicator of the pragmatic distance of sentences. Thus, we design another learning objective. Specifically, we extract the pragmatic features from sentences $s_1$, $s_2$, and $s_3$ and employ the Euclidean distance to measure their pragmatic distances, denoted as $\Delta P^+$ for positive pairs and $\Delta P^-$ for negative pairs:

$$\Delta P^+ = \|\mathbf{h}^p_1 - \mathbf{h}^p_2\|_2, \qquad \Delta P^- = \|\mathbf{h}^p_1 - \mathbf{h}^p_3\|_2. \tag{6}$$

We aim to ensure that sentences in positive pairs exhibit smaller pragmatic distances compared to those in negative pairs. Similar to the implicitness loss, we design a pragmatic loss, $\mathcal{L}_{prag}$, where $\gamma_2$ is a hyperparameter with range $[0, \infty)$.

$$\mathcal{L}_{prag}(s_1, s_2, s_3) = \texttt{max}(0, \gamma_2 - (\Delta P^- - \Delta P^+)). \tag{7}$$

### 3.4 INTEGRATION AND MODEL EMBODIMENT

The final loss function is a weighted sum of $\mathcal{L}_{imp}$ and $\mathcal{L}_{prag}$, where $\alpha$, a predefined factor ranging from $[0, \infty)$, dictates the relative importance assigned to the learning of pragmatics.

$$\mathcal{L}_{final}\big((s_1, s_2), (s_1, s_3)\big) = \mathcal{L}_{imp}(s_1, s_2) + \mathcal{L}_{imp}(s_1, s_3) + \alpha \cdot \mathcal{L}_{prag}(s_1, s_2, s_3). \tag{8}$$

The overall learning objective $\mathcal{J}_\Theta$ is to minimize $\mathcal{L}_{final}$ over all the training set $\mathcal{D}$ in mini-batch approach, where $\Theta = \{\theta, \mathbf{W}_p, \mathbf{W}_s, \mathbf{W}_t\}$ is all the parameters of IMPSCORE:

$$\mathcal{J}_\Theta = \texttt{argmin}_\Theta \sum_{\{(s_1, s_2), (s_1, s_3)\} \in \mathcal{D}_{batch}} \mathcal{L}_{final}\big((s_1, s_2), (s_1, s_3)\big). \tag{9}$$

For model implementation, we embody the Text Encoder with Sentence-BERT (Reimers & Gurevych, 2019) with version `all-mpnet-base-v2`[1]. We prioritize Sentence-BERT over other text encoders, such as BERT (Devlin et al., 2019) and RoBERTa (Liu et al., 2019), because Sentence-BERT is

---

[1] https://sbert.net/docs/sentence_transformer/pretrained_models.html

specifically finetuned to derive semantically meaningful sentence embeddings for sentence pairs, which we believe is beneficial for our task. The maximum input sequence length of Sentence-BERT is 384 and its output dimension is 768. We do not normalize the output embeddings. For the weights of matrices $\mathbf{W}_p$, $\mathbf{W}_s$, and $\mathbf{W}_t$, they are initialized with Xavier Initialization (Glorot & Bengio, 2010): $\mathbf{W}_p$, $\mathbf{W}_s \sim U[-\frac{\sqrt{6}}{\sqrt{d+l}}, \frac{\sqrt{6}}{\sqrt{d+l}}]$, $\mathbf{W}_t \sim U[-\frac{\sqrt{6}}{\sqrt{2l}}, \frac{\sqrt{6}}{\sqrt{2l}}]$.

## 4 TRAINING DATA CONSTRUCTION

To train IMPSCORE, we require dataset of (*implicit sentence*, *explicit sentence*) pairs. However, to the best of our knowledge, no available datasets meet this criteria. Thus, we collected and processed the training data ourselves. We explored a range of existing datasets that indicate implicitness across various NLP tasks, including Textual Entailment/Natural Language Inference, Implicit Hate Speech Detection, Sentiment Analysis, Irony and Sarcasm Detection, and Discourse Relation. Details on dataset construction and processing are provided in Appendix A.2.

Specifically, for implicit hate speech datasets, some already provide explicit explanations for implicit samples, which can be directly used to construct (*implicit*, *explicit*) pairs. For others, we prompted GPT-3.5 with implicit hate speech to generate corresponding explicit ones. Appendix A.1 presents the GPT prompting instructions. This approach was also applied to sentiment analysis datasets. Textual Entailment datasets, which feature sentence pairs (*premise*, *hypothesis*) with entailment relationship "*premise* ⊢ *hypothesis*", can be useful as implicit sentences often entail explicit ones. However, despite the entailment relationship between *premise* and *hypothesis*, not all of them necessarily demonstrate different implicitness levels. For example, in the pair ("*I drove my car yesterday.*", "*I have a car.*"), telling which sentence is more implicit is challenging without additional context and knowing their intention, as both sentences appear explicit. We also present a quality assessment of the constructed data from a linguistic expert in Appendix A.2. In total, we constructed $56,290$ pairs from 13 datasets.

**Positive and Negative Pairs Generation** We treated all $56,290$ pairs constructed above as *positive pairs*, where both sentences are pragmatically aligned. *Negative pairs* were created by replacing the explicit sentence in a positive pair with the explicit sentence from another randomly chosen positive pair within the same dataset. This inner-dataset replacement avoids inconsistencies in implicit-explicit relationships that could arise from the different labeling standards across datasets. Tab. 1 presents the statistics of the training data. Examples of positive and negative pairs are in Tab. 7 in Appendix A.2. We also provide samples of our training data in the Supplementary Material.

Table 1: Statistics of training data. The average length is calculated as the number of characters in a sentence. The number in parentheses is the standard deviation.

| # Positive Pairs | # Negative Pairs | Avg. Length (Implicit Sentences) | Avg. Length (Explicit Sentences) |
|---|---|---|---|
| $56,290$ | $56,290$ | $109.5\ (\pm70.7)$ | $66.94\ (\pm45.5)$ |

## 5 TRAINING OF IMPSCORE

Recall the inference process of IMPSCORE: each input point is in format of $\{(s_1, s_2), (s_1, s_3)\}$ where $s_1$ is an implicit sentence, and $s_2$ and $s_3$ are explicit sentences. $(s_1, s_2)$ denotes the positive pair and $(s_1, s_3)$ for the negative pair. During inference, IMPSCORE computes implicitness scores for $s_1$, $s_2$, and $s_3$, meanwhile, calculates the pragmatic distances for $(s_1, s_2)$ and $(s_1, s_3)$.

### 5.1 EXPERIMENT SETUPS

**Evaluation Metric** We record two results of IMPSCORE's performance during test: 1) *Implicitness Accuracy* — the percentage of IMPSCORE successfully predicting higher implicitness scores for the implicit sentences across all pairs, regardless of whether they are positive or negative. 2) *Pragmatics Accuracy* — the percentage of IMPSCORE successfully predicting smaller pragmatic distances for positive pairs compared to their corresponding negative pairs.

**Hyperparameter Setting** We set the implicitness margin $\gamma_1$ to 0.5, the pragmatic distance margin $\gamma_2$ to 0.7, and $\alpha$ to 1.0. The dimension of two spaces is 64. Adam optimizer is used for gradient descent. The data is randomly split into training, validation, and test sets in an $8:1:1$ ratio. We train for 30 epochs, validating after each epoch, and save the model with the best validation performance on Implicitness Accuracy. The parameter size of IMPSCORE is about 105 MB.

## 5.2 TRAINING PERFORMANCE

Fig. 3 presents the performance of IMPSCORE on test set. The table in left panel shows the Implicitness Accuracy and Pragmatics Accuracy, along with the average implicitness scores for sentences ($I^{imp}$ and $I^{exp}$) and the average pragmatic distances for pairs ($\Delta P^+$ and $\Delta P^-$). IMPSCORE achieves over $95\%$ accuracy in both metrics. The center violin plot illustrates the distribution of implicitness scores of implicit sentences (blue), and for explicit sentences in positive pairs (green) and negative pairs (red). IMPSCORE effectively distinguishes between implicit and explicit ones, demonstrating a distributive bipolarity. The gaps between their average scores are even larger than $\gamma_1 = 0.5$. The right violin plot presents the distribution of pragmatic distances of positive pairs (green) and negative pairs (red). IMPSCORE effectively captures closer pragmatics in positive pairs. Although the output embeddings of IMPSCORE were not normalized, the results suggest that the range of pragmatic distances remains constrained within a narrow section. This is influenced by the properties of embeddings produced by Sentence-BERT. A breakdown of IMPSCORE's testing performance on different data sources is in Appendix A.11.

## 5.3 ABLATION STUDY AND HYPERPARAMETER SENSITIVITY

In this subsection, we examine the effects of different model design and hyperparameter settings. For model design, we focus on two aspects: **distance metric** and **space transformation direction**. We vary the metrics for calculating implicitness scores and pragmatic distances in {Cosine, Euclidean} distances, resulting in four combinations. Besides, we change the space transformation direction in 1) transform pragmatic features to semantic space (p→s); 2) transform semantic features to pragmatic space (s→p); and 3) transform both pragmatic features and semantic features to a third space (p,s→third) using as the follows. $\mathbf{W}_t^1$ and $\mathbf{W}_t^2$ are two independent matrices, and $\mathbf{h}^{tp}$ and $\mathbf{h}^{ts}$ are the transformed pragmatic and semantic features in the third space.

$$I_{\{1,2,3\}} = \left\| \mathbf{h}_{\{1,2,3\}}^{tp} - \mathbf{h}_{\{1,2,3\}}^{ts} \right\|_2, \quad \mathbf{h}_{\{1,2,3\}}^{tp} = \mathbf{h}_{\{1,2,3\}}^{p} \mathbf{W}_t^1, \quad \mathbf{h}_{\{1,2,3\}}^{ts} = \mathbf{h}_{\{1,2,3\}}^{s} \mathbf{W}_t^2 \quad (10)$$

This leads to 3 scenarios. Totally, we have $3 \times 4 = 12$ variations of IMPSCORE. For each variant, we train it three times using the same setting in Sec. 5.1 and calculate the average Implicitness and Pragmatics Accuracy. Their results are summarized in Tab. 2. We observed that using the Cosine metric for calculating implicitness scores yields steadily better results than using Euclidean. However, changes on the metric for calculating pragmatic distances and the space transformation direction do not significantly affect the model's performance.

To assess the impact of hyperparameters, we varied $\gamma_1$ from $\{0.1, 0.3, 0.5, 0.7, 1.0, 1.5\}$, $\gamma_2$ from $\{0.5, 0.7, 1.0, 1.5, 2.0\}$, and $\alpha$ from $\{0.5, 1.0, 1.5, 2.0\}$. We use the original model design of IMPSCORE (i.e. Cosine×Euclidean and p→s). For each hyperparameter setting, we run IMPSCORE 3 times and report the average Implicitness Accuracy. We fixed $\alpha$ and plotted 3D graphs of $\gamma_1$ and $\gamma_2$ on Implicitness Accuracy, smoothing the surface with interpolated points. The result when $\alpha = 1.0$ is presented in Fig. 4, and results of $\alpha = \{0.5, 1.5, 2.0\}$ are in Appendix A.4. We observe that $\gamma_1 = 0.5$ yields the best performance compared to other values. However, IMPSCORE did not show obvious fluctuations to variations in $\gamma_2$ and $\alpha$ within the given ranges. We think the insensitivity

| Implicit Acc. | Pragmatic Acc. |
|---|---|
| 0.952 | 0.962 |
| **Average $I^{imp}$** | **Average $I^{exp}$** |
| 1.296 | 0.514 |
| **Average $\Delta P^+$** | **Average $\Delta P^-$** |
| 0.685 | 1.675 |

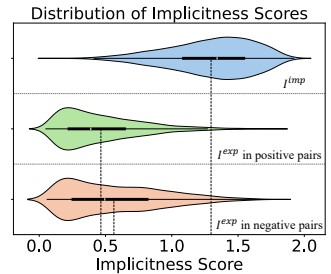
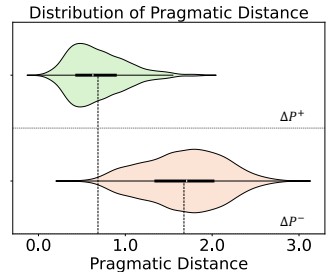

Figure 3: *Left panel*: Detailed results of IMPSCORE on the test set. $I^{imp}$ and $I^{exp}$ denote the implicitness score of implicit sentence and explicit sentence (higher indicates more implicit), and $\Delta P^+$ and $\Delta P^-$ denote the pragmatic distance of positive pair and negative pair (higher indicates pragmatically farther). *Center panel*: The distribution of implicitness scores of implicit sentence, explicit sentence in positive pair, and explicit sentence in negative pair. *Right panel*: The distribution of pragmatic distances of positive pairs and negative pairs.

Table 2: Ablation study results of different model variations of IMPSCORE. The number in the left of / indicates the Implicitness Accuracy, and the right number indicates the Pragmatics Accuracy. Best Implicitness Accuracy are highlighted in **bold**. IMPSCORE is robust to different Space Transformation methods.

| Accuracy (imp/prag) | Space Transformation | | |
|---|---|---|---|
| Metric (imp$\times$prag) | p$\rightarrow$s | s$\rightarrow$p | p,s$\rightarrow$third |
| Cosine$\times$Euclidean | **0.953**/0.962 | 0.947/0.962 | 0.946/0.963 |
| Cosine$\times$Cosine | 0.952/0.973 | **0.953**/0.967 | 0.947/0.971 |
| Euclidean$\times$Cosine | 0.927/0.964 | 0.928/0.974 | 0.929/0.974 |
| Euclidean$\times$Euclidean | 0.926/0.963 | 0.927/0.962 | 0.924/0.963 |

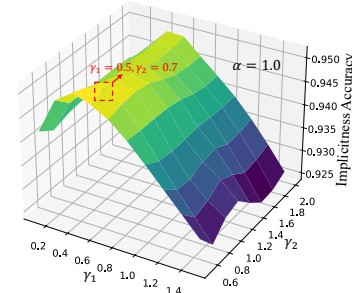

Figure 4: Hyperparameter sensitivity of $\gamma_1$ and $\gamma_2$ on Implicitness Accuracy when $\alpha = 1.0$. Highest point is marked in red dotted box.

to $\gamma_2$ is due to the lack of normalization in the output embeddings and the use of Euclidean distance for calculating pragmatic distance, which enables the model to generate a wide range of scores to optimize the loss function effectively.

Experiments in the following sections all utilized a trained metric, which computes an implicitness score $I$ for each input sentence and an additional pragmatic distance $\Delta P$ for inputted sentence pairs.

## 6 CORRELATION WITH HUMAN JUDGMENTS ON OOD DATA

In this section, we applied the trained IMPSCORE and conducted user study on out-of-distribution (OOD) data to examine the generalization ability of it.

### 6.1 OOD DATA PREPARATION

We prompted GPT-4 with 20 different topics[2] to generate 4 sentences with varying implicitness levels for each topic (*most explicit*, *explicit*, *implicit*, and *most implicit*). The sentences in each topic formed a group (*i.e.*, $G_i$). We, along with a linguist, then manually refined and verified the sentences in each group to ensure they manifest distinct levels of implicitness, allowing us to assign accurate gold ranks (ranked from most explicit to most implicit). Here, we show the user study procedure and results for 10 topics in Tab. 3. Results on the remaining 10 topics are in Appendix A.13. The detailed sentences for each group in Tab. 3 are in Appendix A.3.

### 6.2 USER STUDY DESIGN

We conducted two tasks in the user study. The first task, ***Ranking***, is to rank four sentences within a given topic by their levels of implicitness. It consists of 10 topics ($G_1 - G_{10}$ in Tab. 3). The second task, ***Choice***, is to select the sentence pragmatically closest to a reference sentence from three options. We provided a reference sentence labeled as "explicit" in implicitness level. The three options included one sentence from the same topic group labeled as "most implicit," and two sentences also labeled as "most implicit" but from different topic groups. The *Choice* task also contains 10 questions. See Tab. 9 in details. To avoid fatigue of participant and ensure user study quality, we divide these questions into *Set 1* and *Set 2*[3], where each set contains 5 *Ranking* questions and 5 *Choice* questions. We built user study website on Gorilla and recruited 10 English speakers on Prolific as participants[2]. We divide and assign 5 participants to each set of tasks, so each question has 5 human participants. For testing IMPSCORE, in *Ranking* questions, we compute implicitness scores for 4 sentences and rank them accordingly. In *Choice* questions, we input each option sentence along with the reference sentence into IMPSCORE and calculate their pragmatic distance. The option with smallest pragmatic distance are chosen as the answer. We report Kendall's Tau ($\tau$) and Spearman's Rho ($\rho$) — two metrics for comparing the correlation of two rankings — in *Ranking* task, and report Accuracy in *Choice* task. Kendall's Tau focuses on pairwise concordance, while Spearman's Rho measures the monotonic relationship between rankings. Both $\tau$ and $\rho$ are in range of $[-1, 1]$, where $1$ indicates perfect positive correlation, $-1$ perfect negative correlation, and $0$ no correlation. The calculation of Kendall's Tau is in Eq. 11 in Appendix A.3. Screenshots of the user study questions are in Appendix A.10.

---

[2]Please see the topic design, user study platform, and user participant recruitment details in Appendix A.5.

[3]*Set 1* consists of the first 5 *Ranking* questions in Tab. 8 and the first 5 *Choice* questions in Tab. 9, while *Set 2* consists of the rest.

Table 3: 10 topic groups of OOD data in user study. Design of these topics is in Appendix A.5.

| Group | Topic |
|-------|-------|
| $G_1$ | Ending Relationship |
| $G_2$ | Critiquing a colleague's work |
| $G_3$ | Dealing with a rebellious child |
| $G_4$ | Giving the boss feedback about her behavior |
| $G_5$ | Giving an unfavorable performance review of ballet |
| $G_6$ | Asking a roommate to move out |
| $G_7$ | Handing off a difficult project to a colleague |
| $G_8$ | Disliking John's personality |
| $G_9$ | Decline a friend's party invitation |
| $G_{10}$ | Remind a roommate to clean the kitchen |

Table 4: Inter-participant Kendall's Tau ($\tau$) correlation results in *Ranking* task for user study.

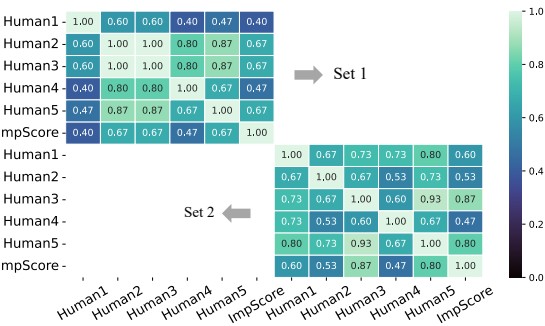

Table 5: Results of participants in *Ranking* and *Choice* tasks compared with **gold ranks/answers** in user study. $\tau$ and $\rho$ stand for Kendall's Tau and Spearman's Rho. Numbers are the average performance of each participant on 5 questions of each task. ↑ indicates that higher results are better.

| Tasks | Human 1 | | Human 2 | | Human 3 | | Human 4 | | Human 5 | | IMPSCORE | |
|-------|---------|---------|---------|---------|---------|---------|---------|---------|---------|---------|---------|---------|
| | Avg. $\tau^\uparrow$ | Avg. $\rho^\uparrow$ | Avg. $\tau^\uparrow$ | Avg. $\rho^\uparrow$ | Avg. $\tau^\uparrow$ | Avg. $\rho^\uparrow$ | Avg. $\tau^\uparrow$ | Avg. $\rho^\uparrow$ | Avg. $\tau^\uparrow$ | Avg. $\rho^\uparrow$ | Avg. $\tau^\uparrow$ | Avg. $\rho^\uparrow$ |
| ***Ranking*** (Set 1) | 0.53 | 0.64 | 0.93 | 0.96 | 0.93 | 0.96 | 0.73 | 0.80 | 0.93 | 0.96 | 0.73 | 0.80 |
| ***Ranking*** (Set 2) | 0.80 | 0.88 | 0.73 | 0.72 | 0.93 | 0.96 | 0.67 | 0.80 | 1.00 | 1.00 | 0.80 | 0.88 |
| | Accuracy | | Accuracy | | Accuracy | | Accuracy | | Accuracy | | Accuracy | |
| ***Choice*** (Set 1) | 0.80 | | 0.80 | | 0.80 | | 0.80 | | 0.80 | | 0.80 | |
| ***Choice*** (Set 2) | 1.00 | | 1.00 | | 0.80 | | 0.60 | | 1.00 | | 1.00 | |

## 6.3 OOD Test Results

The test results of human participants and IMPSCORE on OOD data are in Tab. 4 and Tab. 5, where Tab. 5 details the $\tau$, $\rho$, and Accuracy in comparison to gold labels on both tasks; Tab. 4 shows the inter-participant $\tau$ on the *Ranking* task. Their predictions on each question of both tasks are in Appendix A.3. We observe in Tab. 5 that IMPSCORE achieves satisfactory performances on both *Ranking* and *Choice* tasks compared to the gold labels. Although it does not achieve 100% correctness or match the top performance among human participants, there is no notable performance gap between IMPSCORE and human participant group. The inter-participant $\tau$ results in Tab. 4 show that IMPSCORE has a good correlation with human judgments, with no significant difference from human-human correlation results. Interestingly, sometimes IMPSCORE predicts the same wrong answer as most human participant, suggested in $Q_5$ of *Choice* task in Tab. 9. Results on $\rho$ correlation is presented in Fig. 9. Additionally, we have two exciting observation from IMPSCORE's results:

**Observation 1**: IMPSCORE ranks all "most explicit" (■) and "most implicit" (■) sentences in each *Ranking* question correctly, which can be observed from the results in Tab. 8. Though, IMPSCORE sometimes misranks sentences with minimal differences in implicitness, it performs perfectly when distinguishing between sentences with a significant implicitness gap.

**Observation 2**: IMPSCORE demonstrates consistent and reliable predictions over collections of data. For *Ranking* task, we averaged the implicitness scores of sentences across 10 questions by their implicitness level. The results are presented in Tab. 10. The average scores distinctly reflect the consistent variance across these four levels of sentences. This illustrates that IMPSCORE's performance aligns with the Law of Large Numbers, where despite occasional errors in individual predictions, the average results across many cases converge to the expectation.

## 6.4 Effectiveness of pragmatic and semantic embeddings

In this subsection, we analyzed the effectiveness of the latent pragmatic and semantic embeddings (circles and triangles in Tab. 2) of OOD data in this user study. We retrieved the learned pragmatic and semantic embeddings, and utilized t-SNE to visualize their distributions in 2-D spaces, as displayed in Appendix A.6. The pragmatic distribution in Fig. 11 shows that sentences within the same topic group (expressing the similar meaning) tend to cluster closely, suggesting the effectiveness of their pragmatic embeddings. The semantic distribution in Fig. 12 shows a more dispersed pattern among sentences within the same group. This can be attributed to the diverse semantics used in sentences when expressing the same meaning, which we further illustrate with a case study in Fig. 13. We

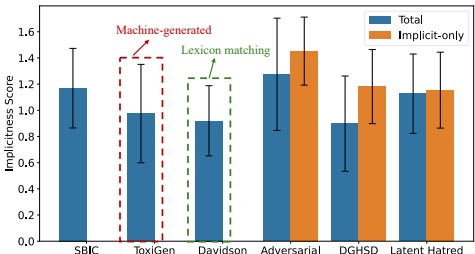
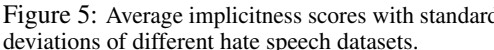
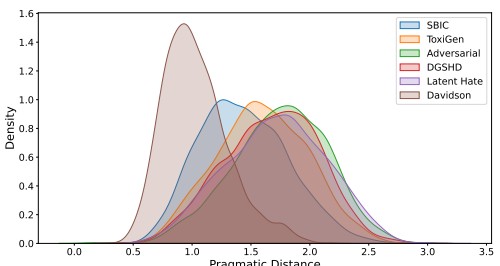

Figure 5: Average implicitness scores with standard deviations of different hate speech datasets.

Figure 6: The pragmatic distances of sentences within each dataset.

selected sentences from two topic groups to analyze their semantic similarities and differences, explaining why some sentences are positioned farther apart than others.

# 7 APPLICATIONS WITH IMPSCORE

## 7.1 IMPLICITNESS LEVELS OF HATE SPEECH BENCHMARKS

The first application is evaluating the implicitness level of different Hate Speech datasets. We selected 6 popular datasets as follows. Note that some already have implicit speech labels.

- ToxiGen (Hartvigsen et al., 2022): A large-scale machine-generated dataset for hate speech detection. It contains $125,672$ hate speeches which are all claimed implicit.
- Latent Hatred (ElSherief et al., 2021): A dataset for understanding implicit hate speech. It contains $8,189$ Twitter posts, among which $7,100$ are labeled as implicit hate speech.
- DGHSD (Vidgen et al., 2021): A dynamically generated dataset to improve online hate detection. It contains $22,175$ hate speeches, among which $3,439$ are implicit ones.
- Adversarial (Ocampo et al., 2023): A syncretic dataset collecting messages from multiple sources. It contains $31,830$ hate speeches, among which $24,823$ are implicit ones.
- SBIC (Sap et al., 2020): A dataset crawling toxic posts from multiple websites like Reddit and Twitter. It contains $24,019$ offensive samples in the training.
- Davidson (Davidson et al., 2017): A dataset crawled from Twitter using lexicon matching. It contains $20,620$ samples that contain hate speech or offensive language.

We processed all their samples through IMPSCORE to infer implicitness scores. For implicit samples already labeled, we recorded their scores separately. Fig. 5 shows the average scores and standard deviations. Implicit samples exhibit higher implicitness levels compared to the corresponding entire datasets. We can see that the Adversarial dataset has the highest implicitness. The Davidson dataset, collected using a lexicon-matching, shows a very low implicitness level. ToxiGen, generated by prompting GPT for implicit hate speech detection, also has low implicitness, potentially indicating the limited ability of LLMs to generate language with implicitness comparable to human levels. Detailed score distributions for each dataset are shown in Fig. 14 in Appendix A.7.

We also examined the pragmatic diversity in each dataset. We randomly sampled $2,000$ distinct sentence pairs from each dataset and calculated their pragmatic distances. The distributions of these distances are shown in Fig. 6. The Davidson dataset exhibits a notably lower pragmatic distance compared to others, indicating the least diversity in sentence pragmatics.

These findings demonstrate how IMPSCORE can be used to compare the properties of different datasets, offering valuable insights for selecting appropriate benchmarks in experimental studies.

## 7.2 LANGUAGE MODELS' CAPABILITIES ON DIFFERENT IMPLICITNESS LEVELS

In the second application, we aim to explore LLMs' understanding capabilities of language with different levels of implicitness. We selected two state-of-the-art LLMs: GPT-4-Turbo and Llama-3.1-8B-Instruct, along with a content moderation tool — OpenAI Moderation[4]. We chose three mentioned datasets: ToxiGen, DGHSD, and Adversarial, as our test sets. The first two demonstrate a wide spectrum of implicitness, while Adversarial predominantly contains samples with higher levels of implicitness, as depicted in Fig. 14 in Appendix. We divided their samples into 8 subsets based on their implicitness scores, ranging from $[0, 0.25)$ to $[1.75, 2]$, with each interval spanning

---

[4]https://platform.openai.com/docs/guides/moderation

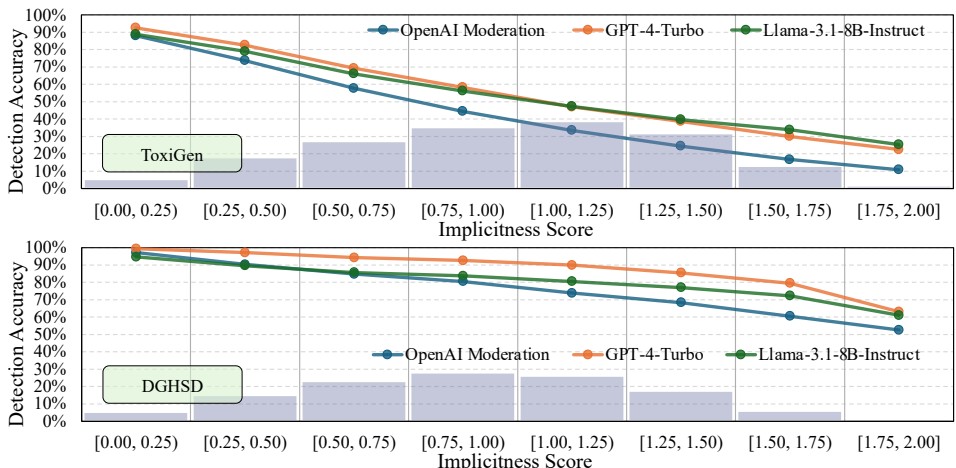

Figure 7: Hate speech detection accuracy of LLMs on sentence samples with different levels of implicitness. Blue columns indicate the distribution of samples among different ranges of implicitness.

0.25. The task is a binary classification where we present hate speech to LLMs and ask them to determine whether it is problematic in zero-shot prompting scenario. The prompt format is presented in Tab. 15 in Appendix A.8. For OpenAI Moderation, we applied a lenient criterion by checking if it flags content as potentially harmful, without necessarily categorizing it as hate speech. We report Accuracy, the proportion of test samples that LLMs successfully identify as problematic.

The detection accuracy, displayed in Fig. 7 for ToxiGen and DGHSD and Fig. 16 in Appendix for Adversarial A.9, reveal that detection accuracy declines as the implicitness level of sentences increases. In ToxiGen and DGHSD, accuracy for the three LLMs drops consistently and they have very low detection success rates in high implicitness ranges. In Adversarial, GPT-4-Turbo and Llama-3.1-8B-Instruct show performance improvement on range $[1.75, 2]$. We attribute this to the samples in this range being very similar, reducing diversity and potentially biasing the results, as demonstrated in Appendix A.9. Overall, the results accord with our expectation, underscoring the effectiveness of IMPSCORE. The clear downward trend in performance highlights a significant limitation in LLMs' understanding of highly implicit sentences. This problem is statistically undiscovered in the past and also overlooked by the high average benchmark performances (Jahan & Oussalah, 2023).

Additionally, we applied IMPSCORE to assess the generation capabilities of different LLMs, as detailed in Appendix A.15.

## 8  CONCLUSION AND DISCUSSION

In this paper, we addressed the problem of quantifying the implicitness level of language. By providing an interpretable definition of "implicitness," we developed a metric capable of computing implicitness scores for sentences. IMPSCORE's performance on out-of-distribution test data demonstrates satisfactory accuracy and alignment with human judgments, which highlights its generalizability.

However, this research has limitations that suggest directions for future work. First, while the dataset comprising $112,580$ sentence pairs, expanding it could improve IMPSCORE's accuracy and generalizability. To do so, beyond expanding the search for existing datasets, leveraging LLMs to generate synthetic data while maintaining a coherent speech style is a promising approach. Moreover, incorporating a broader range of implicitness types and multilingual data would enhance IMPSCORE's versatility. Second, IMPSCORE's evaluation, though comprehensive, could benefit from further user studies and a wider range of inventory questions. Third, as the dataset grows, employing more advanced text encoders like SimCSE (Gao et al., 2021) may further improve performance. To this end, we encourage the research community to explore and refine IMPSCORE's capabilities.

Looking forward, IMPSCORE has broad potential in NLP. It could assess implicitness in machine-generated text, aiding both retrospective analysis and model training. Specifically, IMPSCORE could serve as a tool in reinforcement learning setups, providing a reward signal that enhances language models' ability to handle nuanced linguistic constructs. Additionally, IMPSCORE's capacity for screening complex, high-implicitness data could significantly benefit tasks in intent detection, social awareness, and theory of mind analyses, among others.

ACKNOWLEDGEMENTS

This research was supported in part by a grant from the US National Library of Medicine (R01LM013833).

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

## A    APPENDIX

### A.1    GPT PROMPTS FOR POSITIVE PAIR GENERATION

> **role**: system
> **content** (*ToxiGEN*): Generate a more explicit sentence that the given hate speech implies. For example, "it is not a good idea to be trustful of foreigners." → "foreigners are not trustful." Don't put any prefix like "Explicit version:" in the output.
>
> **content** (*DGHSD*): Generate a more explicit sentence that the given hate speech implies. For example, "it is not a good idea to be trustful of foreigners." → "foreigners are not trustful." Don't put any prefix like "Explicit version:" in the output.
>
> **content** (*Adversarial*): Generate a more explicit sentence that the given hate speech implies. For example, "it is not a good idea to be trustful of foreigners." → "foreigners are not trustful." Don't put any prefix like "Explicit version:" in the output.
>
> **content** (*Implicatures*): In the given conversation, what does the response imply? Please directly generate an explicit version of the response. Don't put "An explicit version:" or " <Explicit Response>:" in the output.
>
> **content** (*LUDWIG*): In the given conversation between A and B, what does B's response imply? Please directly generate an explicit version of B's response. Don't put "B:" or "An explicit version:" such kinds of prefix in the output.
>
> **content** (*ABSA*): Generate a more explicit sentence the given sentence implies. Don't put any prefix like "Explicit version:" in the output.
>
> **content** (*SemEval*): The given sentence contains irony or sarcasm. Please generate an more explicit version of the sentence expressing the actual meaning. Don't put any prefix like "Explicit version:" in the output.
>
> **content** (*Snarks*): The given sentence is sarcastic. Please generate an explicit version of the sentence expressing the actual meaning. Don't put any prefix like "Explicit version:" in the output.
> **role**: user
> **content**: <Inserted Input Implicit Sentence>

Figure 8: Prompts for generating positive pairs using GPTs from different data sources.

### A.2    TRAINING DATA DETAILS

We constructed training data from existing public data sources over various NLP research topics. These sources include ToxiGEN (Hartvigsen et al., 2022), Latent Hatred (ElSherief et al., 2021), Dynamically-Generated-Hate-Speech-Dataset (DGHSD) (Vidgen et al., 2021), Adversarial-Implicit-Hate-Speech (Adversarial) (Ocampo et al., 2023), SBIC (Sap et al., 2020) from the topic of hate speech detection; Implicatures (Sileo, 2023), ImpPres (Jeretic et al., 2020), LUDWIG (George & Mamidi, 2020) from the topic of natural language inference; ABSA (Li et al., 2021) from the topic of Sentiment Analysis; SemEval-2018 Task 3 dataset (Rohanian et al., 2018) from the topic of Irony Detection; Snarks (Srivastava et al., 2023) and R³(Chakrabarty et al., 2020) from the topic of Sarcasm Detection; and PDTB (Prasad et al., 2008; Tonelli & Cabrio, 2012) from the discourse relation study. Tab. 6 summarizes the processing details.

Table 6: Processing details of our training data construction. The column **# Pos. Pairs** represents the number of positive (*implicit*, *explicit*) training pairs constructed from each data source. The column **Generation** describes the method used to construct these pairs, either through prompts to GPT models or by using original labels annotated in the data sources.

| Data Source | # Pos. Pairs | Generation | Processing Description |
|---|---|---|---|
| **Implicit Hate Speech Detection** | | | |
| ToxiGEN | 2,648 | GPT-3.5[5] | Each hate speech instance in ToxiGEN is rated for "toxic level" (1.0 to 5.0) and "intent level" (1.0 to 5.0). We selected sentences with toxic levels from 2.0 to 4.0 and intent levels below 4.0 as implicit, and those with both levels above 4.0 as explicit. We prompted GPT with implicit sentences to generate positive pairs. |
| Latent Hatred | 12,036 | Original | This dataset contains Twitter posts labeled as "implicit hatred" or "explicit hatred". The implicit posts are further explained to clarify what they imply. We pair these implicit posts with their explanations to obtain positive pairs. |
| DGHSD | 3,415 | GPT-3.5 | Some hate speech are labeled as "implicit". We prompt GPT with implicit sentences to generate positive pairs. |
| Adversarial | 25,193 | GPT-3.5 | Some hate speech is labeled as "implicit" and some is labeled as "explicit". We prompt GPT with implicit sentences to generate positive pairs. |
| SBIC | 8,316 | Original | Some sentence samples are explained to clarify why they are implicitly biased. We pair them with their explanations to create positive pairs. |
| **Natural Language Inference / Textual Entailment** | | | |
| Implicatures | 955 | GPT-3.5 | Each sample is a dialogue containing a context and an original response. The original response is an implicit sentence. We prompt GPT with the dialogue to generate an explicit response. We then pair the implicit sentences and explicit ones to get positive pairs. |
| IMPPRES | 300 | Original | We use the following three files of the implicature part of the IMPPRES dataset: `gradable_adjective.jsonl`, `gradable_verb.jsonl`, `modals.jsonl` where each data point consists of a sentence and its implicature, corresponding to an implicit sentence and a explicit sentence respectively. |
| LUDWIG | 718 | GPT-3.5 | As a part of BIG-bench, LUDWIG is a dataset of conversational implicatures where each utterance (question) is associated with a conversational response (implicit answer) and its implicature (explicit answer; i.e. yes/no). We append questions with implicit answers to create implicit sentences and questions with explicit answers to create explicit sentences. |
| **Sentiment Analysis** | | | |
| ABSA | 933 | GPT-3.5 | Each sample is an independent sentence, and some are labeled as "implicit sentiment" or "non-implicit sentiment". We prompt GPT with implicit ones and get positive pairs. |
| **Irony / Sarcasm Detection** | | | |
| SemEval | 1,316 | GPT-4[5] | Some tweet posts are labeled as ironic. We prompt GPT with implicit ones and get positive pairs. We found that GPT-3.5 models underperform in understanding irony, so we switch to GPT-4 instead. |
| Snarks | 180 | GPT-4 | This datasets contains sentence pairs which are lexical similar. In each pair, one is a sarcasm and the other is not. We prompt GPT with sarcasm ones and get positive pairs. |
| R$^3$ | 150 | Original | This dataset contains sarcasm sentences with which we prompt GPT to obtain positive pairs. |
| **Discourse Relation** | | | |
| PDTB | 130 | Original | Each sample is a hypothesis pair (*Text → Hypothesis*), which has been manually annotated as as entailing or not entailing. We further convert them to (*implicit*, *explicit*) positive pairs. |
| **Total** | **56,290** | | |

---

[5]Specifically, we use GPT-3.5-turbo-0125 and GPT-4o models.

Our constructed dataset has also been assessed by a linguistic expert. 20 sentence pairs (10 positive and 10 negative) were randomly sampled from each source for expert annotation, and an integer from 1 to 5 is given to each dataset using the following metric: For a given sampled sentence pair, it is considered appropriate if (1) it is a positive pair where the two sentences are the same in the pragmatic dimension and follows the (*implicit sentence*, *explicit sentence*) pattern or (2) it is a negative pair where the two sentences are pragmatically distinct but also follow the (*implicit sentence*, *explicit sentence*) pattern. A fitness integer $s$ ranging from **1** to **5** is assigned to each source based on the percentage of sampled sentences that fit our design goal: **1** – $s < 20\%$; **2** – $20\% \leq s < 40\%$; **3** – $40\% \leq s < 60\%$; **4** – $60\% \leq s < 80\%$; **5** – $s \geq 80\%$. The quality assessments, along with examples of positive and negative pairs, are presented in Tab. 7.

Table 7: Quality assessment and examples of positive and negative pairs of our constructed training data. **Note that some examples are highly toxic and harmful**. A linguistic expert rated samples from each source dataset on a scale from 1 to 5.

| Data Source | Quality Rating | Positive and Negative Pair Example (*implicit sentence*, *explicit sentence*) |
|---|---|---|
| | | **Implicit Hate Speech Detection** |
| ToxiGEN | 5/5 | **Pos**: ("the British were not being fair when they played cards with the native americans", "Native Americans are not capable of playing cards fairly.") **Neg**: ("the British were not being fair when they played cards with the native americans", "if youre not white youre not american.") |
| Latent Hatred | 5/5 | **Pos**: ("if you are white in that state watch yourself !", "non-white people are dangerous.") **Neg**: ("if you are white in that state watch yourself !", "Minorities are inferior to whites.") |
| DGHSD | 5/5 | **Pos**: ("I am pretty glad I was born white", "White people are superior.") **Neg**: ("I am pretty glad I was born white", "Most of us aren't yellow fever idiots though and respect your entire culture. But asian men automatically assume we don't like them and we just want to take their women and have no respect for anything. That's not true at all, we just don't respect them.") |
| Adversarial | 5/5 | **Pos**: ("what has feminism ever done for you ?", "Feminism has not benefited you in any way.") **Neg**: ("what has feminism ever done for you ?", "the creation story is proof that the native americans were no civilized society in fact one of the most savage peoples to ever walk the earth.") |
| SBIC | 5/5 | **Pos**: ("What runs in every black family? The dad", "black men always leave their families.") **Neg**: ("What runs in every black family? The dad", "makes it souns as if jewish people are not part of the white race.") |
| | | **Natural Language Inference / Textual Entailment** |
| Implicatures | 4/5 | **Pos**: ("- Do you expect Sheila to be late for the party tonight? - Is the pope Catholic?", "- Do you expect Sheila to be late for the party tonight? - Yes, Sheila is always late for everything.") **Neg**: ("- Do you expect Sheila to be late for the party tonight? - Is the pope Catholic?", "- Are you going to keep your part time job the next semest at the biology laboratory? - You do not plan to keep your part-time job at the biology laboratory for the next semester.") |
| IMPPRES | 5/5 | **Pos**: ("The ladders are fine.", "The ladders are not great.") **Neg**: ("The ladders are fine.", "Kristin did not manage to confuse Lisa.") |
| LUDWIG | 5/5 | **Pos**: ("- Do you like my new outfit? - You shouldn't be allowed to buy clothes.", "- Do you like my new outfit? - That outfit does not look good on you.") **Neg**: ("- Do you like my new outfit? - You shouldn't be allowed to buy clothes.", "- Are you going to talk to Mark? - I need to talk to Mark and address something that has been bothering me.") |
| | | **Sentiment Analysis** |
| ABSA | 4/5 | **Pos**: ("I didn't go there for food so I can't comment.", "Since I did not go to that place specifically to eat, I do not have any feedback or opinions on the food available there.") **Neg**: ("I didn't go there for food so I can't comment.", "One drawback, I wish the keys were backlit.") |
| | | **Irony / Sarcasm Detection** |

| | | |
|---|---|---|
| SemEval | 5/5 | **Pos**: ("Sweet United Nations video. Just in time for Christmas.", "The United Nations released a video, but it seems poorly timed or irrelevant for the Christmas season.") **Neg**: ("Sweet United Nations video. Just in time for Christmas.", "THEY BE should ALLOW more REFUGEES among them will be potential.") |
| Snarks | 5/5 | **Pos**: ("Nobody can be successful without having inherited all their money first.", "People can achieve success through their own hard work and determination, without needing to inherit money.") **Neg**: ("Nobody can be successful without having inherited all their money first.", "The best way to defeat a stronger/larger and better regimented force is by fighting on your terms.") |
| $R^3$ | 5/5 | **Pos**: ("Mom is in a good mood today. She is just old and cranky.", "Mom is in a bad mood today.") **Neg**: ("Mom is in a good mood today. She is just old and cranky.", "Such an annoying start for my morning.") |
| *Discourse Relation* | | |
| PDTB | 4/5 | **Pos**: ("Look how much air is moving around.", "The ventilation here is great.") **Neg**: ("Look how much air is moving around.", "Investors are buying stocks that have predictable earnings.") |

### A.3 USER STUDY QUESTIONS AND RESULTS

Tab. 8 presents the detailed questions in the *Ranking* task and ranks answered by human participants, IMPSCORE, and the gold ranks. Tab. 9 presents the detailed questions in the *Choice* task and answers of human participants, IMPSCORE, and gold answers. Colored boxes are associated with specific sentence options in each question.

Table 8: Details of the 10 questions in *Ranking* task of the user study, along with answers from human participants, IMPSCORE, and the gold rank for each question.

| $Q_1$ / Group $G_1$ | **Topic: Ending Relationship** |
|---|---|

■ I've made up my mind to end this relationship. It's clear as day that we need to go our separate ways.
■ It seems to me that our association has reached its natural conclusion.
■ Our recent challenges have made me think about our future. Do you think we're really right for each other?
■ I've been reflecting a lot on happiness and fulfillment, both for myself and the people I care about. What are your thoughts on where we stand?

| Rankings | Human1 | Human2 | Human3 | Human4 | Human5 | IMPSCORE | Gold Rank |
|---|---|---|---|---|---|---|---|
| most explicit | ■ | ■ | ■ | ■ | ■ | ■ | ■ |
| explicit | ■ | ■ | ■ | ■ | ■ | ■ | ■ |
| implicit | ■ | ■ | ■ | ■ | ■ | ■ | ■ |
| most implicit | ■ | ■ | ■ | ■ | ■ | ■ | ■ |

| $Q_2$ / Group $G_2$ | **Topic: Critiquing a colleague's work** |
|---|---|

■ This report has several inaccuracies and the analysis lacks depth. We need to address these errors immediately.
■ There are a few points in your report that aren't aligning with our data standards. We should revise these sections for accuracy.
■ Your report offers some interesting insights, but I wonder if a deeper data review could enhance our conclusions.
■ Your report has sparked interesting thoughts about our project. Maybe we could review the data together to explore other potential directions?

| Rankings | Human1 | Human2 | Human3 | Human4 | Human5 | IMPSCORE | Gold Rank |
|---|---|---|---|---|---|---|---|
| most explicit | ■ | ■ | ■ | ■ | ■ | ■ | ■ |
| explicit | ■ | ■ | ■ | ■ | ■ | ■ | ■ |
| implicit | ■ | ■ | ■ | ■ | ■ | ■ | ■ |
| most implicit | ■ | ■ | ■ | ■ | ■ | ■ | ■ |

$Q_3$ / Group $G_3$   **Topic: Dealing with a rebellious child**

🟥 You need to follow the rules of our house or there will be consequences.
🟩 We must understand, my child, that respecting our household rules is crucial.
🟦 Do you think we could talk about your recent behavior and how it fits with our family expectations?
🟧 Do you think we're on the same page about how things should be at home?

| Rankings | Human1 | Human2 | Human3 | Human4 | Human5 | IMPSCORE | Gold Rank |
|---|---|---|---|---|---|---|---|
| most explicit | teal | magenta | magenta | magenta | magenta | magenta | magenta |
| explicit | magenta | teal | teal | teal | teal | blue | teal |
| implicit | blue | blue | blue | orange | blue | orange | blue |
| most implicit | orange | orange | orange | blue | orange | teal | orange |

$Q_4$ / Group $G_4$   **Topic: Giving the boss feedback about her behavior**

🟥 Your recent behavior has negatively affected our team's dynamics and needs to be addressed immediately.
🟩 I've noticed some actions from you that could be impacting team morale negatively.
🟦 I think exploring different approaches to team management might benefit us all. What are your thoughts?
🟧 Have you considered the impact of our leadership styles on the team's atmosphere?

| Rankings | Human1 | Human2 | Human3 | Human4 | Human5 | IMPSCORE | Gold Rank |
|---|---|---|---|---|---|---|---|
| most explicit | magenta | magenta | magenta | magenta | magenta | magenta | magenta |
| explicit | teal | teal | teal | teal | teal | blue | teal |
| implicit | orange | orange | orange | orange | blue | teal | blue |
| most implicit | blue | blue | blue | blue | orange | orange | orange |

$Q_5$ / Group $G_5$   **Topic: Giving an unfavorable performance review of ballet**

🟥 The performance did not meet the expected standards of technique and expression.
🟩 Several sections require substantial revisions to ensure they fully capture the artistic intent and emotional depth required.
🟦 The ballet might benefit from additional practice in precision and synchrony.
🟧 The performance hinted at classical roots, yet further rehearsals could enhance its clarity and cohesion.

| Rankings | Human1 | Human2 | Human3 | Human4 | Human5 | IMPSCORE | Gold Rank |
|---|---|---|---|---|---|---|---|
| most explicit | teal | magenta | magenta | magenta | magenta | teal | magenta |
| explicit | magenta | teal | teal | blue | teal | magenta | teal |
| implicit | blue | blue | blue | orange | blue | blue | blue |
| most implicit | orange | orange | orange | teal | orange | orange | orange |

$Q_6$ / Group $G_6$   **Topic: Asking a roommate to move out**

🟥 You must find a new place and move out by the end of this month.
🟩 Our living arrangement isn't working anymore. It's time for you to find a new place.
🟦 Should we consider other living situations to ease tensions?
🟧 Maybe exploring other housing options could benefit us both?

| Rankings | Human1 | Human2 | Human3 | Human4 | Human5 | IMPSCORE | Gold Rank |
|---|---|---|---|---|---|---|---|
| most explicit | magenta | magenta | magenta | magenta | magenta | teal | magenta |
| explicit | teal | teal | teal | teal | teal | magenta | teal |
| implicit | blue | blue | blue | orange | blue | blue | blue |
| most implicit | orange | orange | orange | blue | orange | orange | orange |

$Q_7$ / Group $G_7$   **Topic: Handing off a difficult project to a colleague**

🟥 This project has become too difficult for me. I think your skills are better suited for it. Can you take it over?
🟩 I'm struggling with this project and believe you have the right skills to manage it. Would you be willing to take over?
🟦 You have a flair for managing complex challenges. This project is proving tough for me. Interested in taking the reins?
🟧 I've seen how well you handle tricky projects. There's one on my desk that seems right up your alley. Fancy giving it a whirl?

| Rankings | Human1 | Human2 | Human3 | Human4 | Human5 | IMPSCORE | Gold Rank |
|---|---|---|---|---|---|---|---|

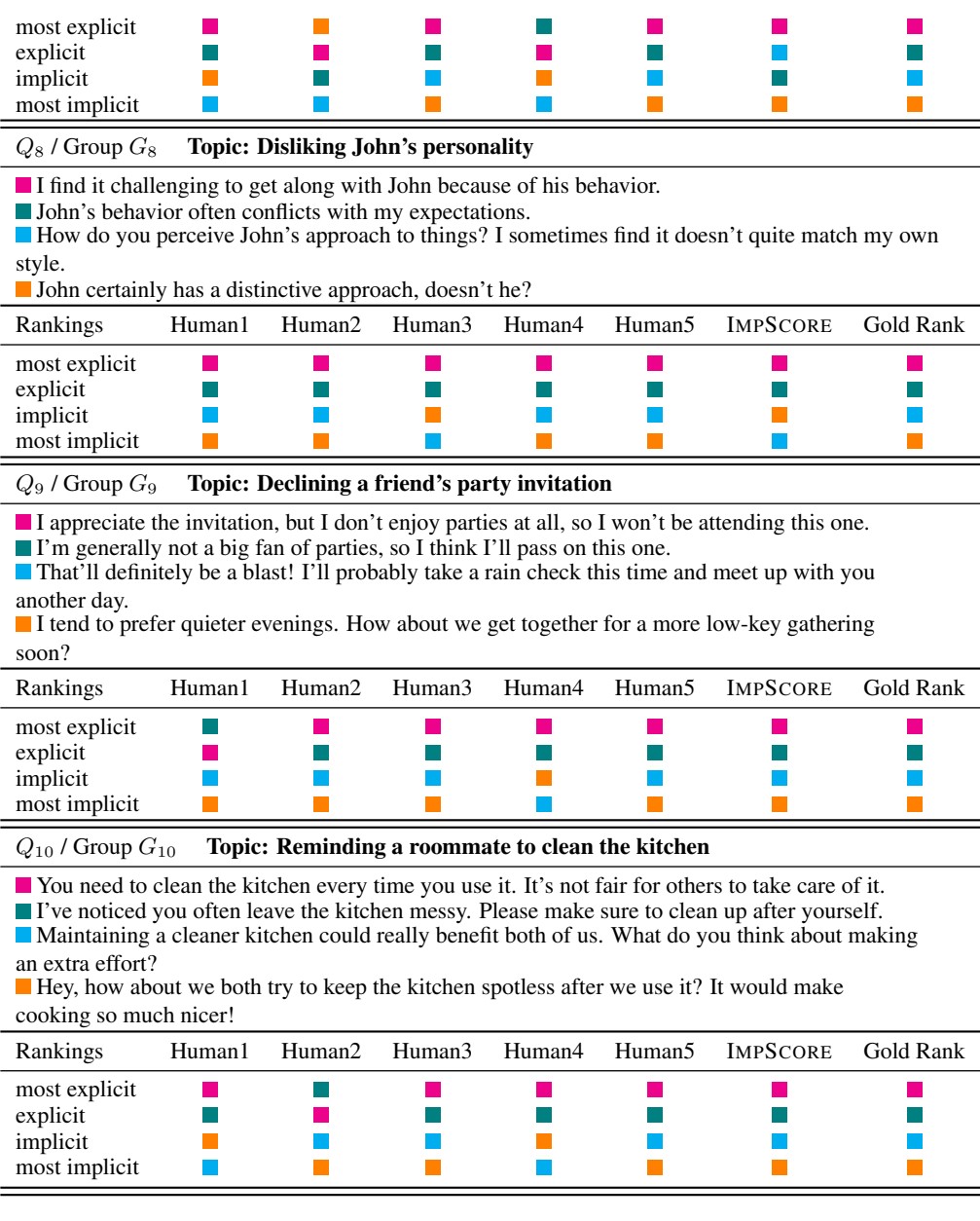

| Rankings | Human1 | Human2 | Human3 | Human4 | Human5 | IMPSCORE | Gold Rank |
|---|---|---|---|---|---|---|---|
| most explicit | | | | | | | |
| explicit | | | | | | | |
| implicit | | | | | | | |
| most implicit | | | | | | | |

$Q_8$ / Group $G_8$     **Topic: Disliking John's personality**

■ I find it challenging to get along with John because of his behavior.
■ John's behavior often conflicts with my expectations.
■ How do you perceive John's approach to things? I sometimes find it doesn't quite match my own style.
■ John certainly has a distinctive approach, doesn't he?

$Q_9$ / Group $G_9$     **Topic: Declining a friend's party invitation**

■ I appreciate the invitation, but I don't enjoy parties at all, so I won't be attending this one.
■ I'm generally not a big fan of parties, so I think I'll pass on this one.
■ That'll definitely be a blast! I'll probably take a rain check this time and meet up with you another day.
■ I tend to prefer quieter evenings. How about we get together for a more low-key gathering soon?

$Q_{10}$ / Group $G_{10}$     **Topic: Reminding a roommate to clean the kitchen**

■ You need to clean the kitchen every time you use it. It's not fair for others to take care of it.
■ I've noticed you often leave the kitchen messy. Please make sure to clean up after yourself.
■ Maintaining a cleaner kitchen could really benefit both of us. What do you think about making an extra effort?
■ Hey, how about we both try to keep the kitchen spotless after we use it? It would make cooking so much nicer!

Table 9: Details of the 10 questions in *Choice* task of the user study, along with answers from human participants, IMPSCORE, and the gold answer for each question.

| $Q_1$ **Reference sentence**: I noticed a few unusual behaviors during the exam that I'd like to understand better. | | | | | | | |
|---|---|---|---|---|---|---|---|

■ Exams can be overwhelming, and sometimes they might prompt actions we wouldn't normally consider.
■ Perhaps the new team member would thrive with a bit more mentoring on our specific methods.
■ I've seen how well you handle tricky projects. There's one on my desk that seems right up your alley. Fancy giving it a whirl?

| Answer | Human1 | Human2 | Human3 | Human4 | Human5 | IMPSCORE | Gold Answer |
|---|---|---|---|---|---|---|---|
| Closest one | ■ | ■ | ■ | ■ | ■ | ■ | ■ |

$Q_2$ **Reference sentence**: Hey, the scent of your dinner is pretty powerful. What do you think about eating it outside?

- ■ (pink) Your dinner smells intense! Maybe it would be more enjoyable if you had it outside in the fresh air?
- ■ (blue) Do you think we're on the same page about how things should be at home?
- ■ (orange) You really savor your noodles loudly! It's interesting, as it's quite the opposite in my culture, where we eat quietly.

| Answer | Human1 | Human2 | Human3 | Human4 | Human5 | IMPSCORE | Gold Answer |
|---|---|---|---|---|---|---|---|
| Closest one | ■ (pink) | ■ (pink) | ■ (pink) | ■ (pink) | ■ (pink) | ■ (pink) | ■ (pink) |

$Q_3$ **Reference sentence**: You have a flair for managing complex challenges. This project is proving tough for me. Interested in taking the reins?

- ■ (pink) I've seen how well you handle tricky projects. There's one on my desk that seems right up your alley. Fancy giving it a whirl?
- ■ (blue) Have you considered the impact of our leadership styles on the team's atmosphere?
- ■ (orange) Reflecting on my optimal working conditions, I'm curious if less oversight might enhance my output.

| Answer | Human1 | Human2 | Human3 | Human4 | Human5 | IMPSCORE | Gold Answer |
|---|---|---|---|---|---|---|---|
| Closest one | ■ (pink) | ■ (pink) | ■ (pink) | ■ (pink) | ■ (pink) | ■ (pink) | ■ (pink) |

$Q_4$ **Reference sentence**: It's a bit tight for me financially these days.

- ■ (pink) Mixing money and friendship often blurs the lines.
- ■ (blue) John certainly has a distinctive approach, doesn't he?
- ■ (orange) How do you think you could boost your contribution to our team?

| Answer | Human1 | Human2 | Human3 | Human4 | Human5 | IMPSCORE | Gold Answer |
|---|---|---|---|---|---|---|---|
| Closest one | ■ (pink) | ■ (orange) | ■ (pink) | ■ (pink) | ■ (orange) | ■ (pink) | ■ (pink) |

$Q_5$ **Reference sentence**: Could we perhaps look at other options that might align more closely with our expectations?

- ■ (pink) Is there a chance we could turn over a new leaf with a different model or terms?
- ■ (blue) Hey, how about we both try to keep the kitchen spotless after we use it? It would make cooking so much nicer!
- ■ (orange) Your report has sparked interesting thoughts about our project. Maybe we could review the data together to explore other potential directions?

| Answer | Human1 | Human2 | Human3 | Human4 | Human5 | IMPSCORE | Gold Answer |
|---|---|---|---|---|---|---|---|
| Closest one | ■ (orange) | ■ (pink) | ■ (orange) | ■ (orange) | ■ (pink) | ■ (orange) | ■ (pink) |

$Q_6$ **Reference sentence**: That restaurant is not exactly my favorite — could we look at some other options?

- ■ (pink) That restaurant is a curious choice. Perhaps we could check out a few other places as well?
- ■ (blue) Your dinner smells intense! Maybe it would be more enjoyable if you had it outside in the fresh air?
- ■ (orange) You really savor your noodles loudly! It's interesting, as it's quite the opposite in my culture, where we eat quietly.

| Answer | Human1 | Human2 | Human3 | Human4 | Human5 | IMPSCORE | Gold Answer |
|---|---|---|---|---|---|---|---|
| Closest one | ■ (pink) | ■ (pink) | ■ (pink) | ■ (pink) | ■ (pink) | ■ (pink) | ■ (pink) |

$Q_7$ **Reference sentence**: I've been thinking about how I work best — with more freedom in handling my tasks.

- ■ (pink) Reflecting on my optimal working conditions, I'm curious if less oversight might enhance my output.
- ■ (blue) Could we discuss our financial arrangement? It's time to sort out a plan.
- ■ (orange) Hey, how about we both try to keep the kitchen spotless after we use it? It would make cooking so much nicer!

| Answer | Human1 | Human2 | Human3 | Human4 | Human5 | IMPSCORE | Gold Answer |
|---|---|---|---|---|---|---|---|
| Closest one | ■ (pink) | ■ (pink) | ■ (pink) | ■ (pink) | ■ (pink) | ■ (pink) | ■ (pink) |

$Q_8$ **Reference sentence**: Do you think we could try a new approach where we handle more things by ourselves?

- ■ (pink) We value your advice, but we're keen on handling things ourselves to learn and grow.
- ■ (blue) Perhaps the new team member would thrive with a bit more mentoring on our specific methods.
- ■ (orange) How do you think you could boost your contribution to our team?

| Answer | Human1 | Human2 | Human3 | Human4 | Human5 | IMPSCORE | Gold Answer |
|---|---|---|---|---|---|---|---|
| Closest one | 🟥 | 🟥 | 🟦 | 🟦 | 🟥 | 🟥 | 🟥 |

$Q_9$ **Reference sentence**: Have you tried the new deodorants? They're great for staying fresh all day.

🟥 I just found some amazing products for staying fresh longer. I think you might like them too.
🟦 Do you think we're on the same page about how things should be at home?
🟧 Hey, how about we both try to keep the kitchen spotless after we use it? It would make cooking so much nicer!

| Answer | Human1 | Human2 | Human3 | Human4 | Human5 | IMPSCORE | Gold Answer |
|---|---|---|---|---|---|---|---|
| Closest one | 🟥 | 🟥 | 🟥 | 🟧 | 🟥 | 🟥 | 🟥 |

$Q_{10}$ **Reference sentence**: Do you think we could talk about your recent behavior and how it fits with our family expectations?

🟥 Do you think we're on the same page about how things should be at home?
🟦 The performance hinted at classical roots, yet further rehearsals could enhance its clarity and cohesion.
🟧 Maybe exploring other housing options could benefit us both?

| Answer | Human1 | Human2 | Human3 | Human4 | Human5 | IMPSCORE | Gold Answer |
|---|---|---|---|---|---|---|---|
| Closest one | 🟥 | 🟥 | 🟥 | 🟥 | 🟥 | 🟥 | 🟥 |

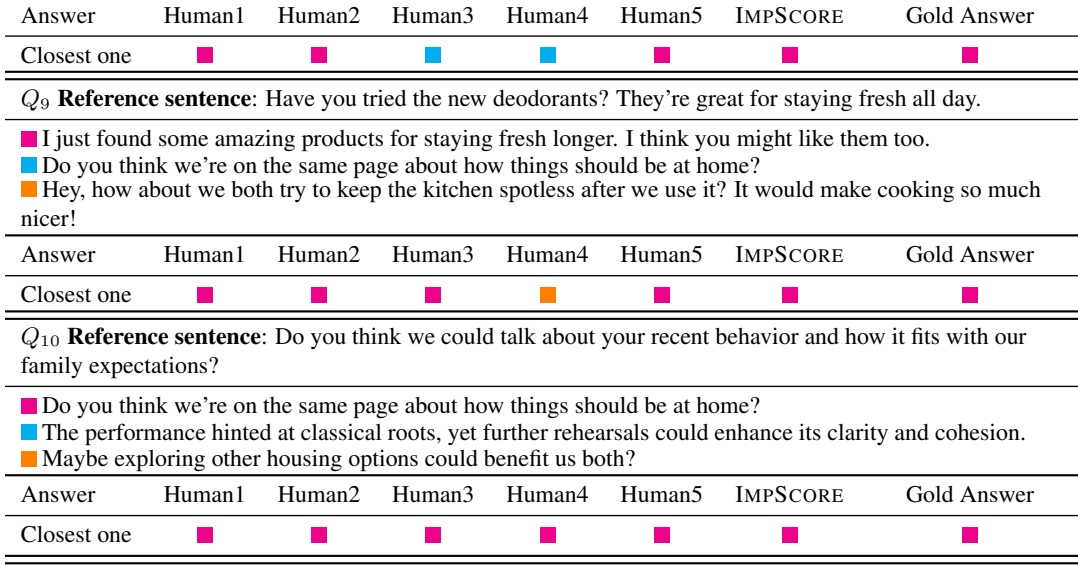

Figure 9: Inter-participant Spearman's Rho ($\rho$) correlation results in *Ranking* task for user study.

Table 10: Implicitness scores computed by IMPSCORE for 4 sentences in 10 *Ranking* task questions. The last row indicates the average scores across questions of 4 levels sentences. Higher scores indicate higher implicitness levels.

| Topic Group | Implicitness score of each sentence computed by IMPSCORE | | | |
|---|---|---|---|---|
| | Most explicit sentence | Explicit sentence | Implicit sentence | Most implicit sentence |
| $G_1$ | 0.91 | 0.96 | 1.10 | 1.55 |
| $G_2$ | 0.94 | 0.96 | 1.10 | 1.18 |
| $G_3$ | 0.90 | 0.66 | 0.87 | 1.52 |
| $G_4$ | 0.44 | 0.67 | 0.57 | 0.97 |
| $G_5$ | 0.22 | 0.72 | 0.88 | 0.83 |
| $G_6$ | 0.93 | 0.94 | 1.50 | 1.36 |
| $G_7$ | 0.53 | 0.89 | 0.86 | 1.30 |
| $G_8$ | 0.49 | 0.33 | 1.04 | 1.40 |
| $G_9$ | 0.67 | 1.40 | 1.57 | 1.73 |
| $G_{10}$ | 0.90 | 0.91 | 1.13 | 1.84 |
| | **Average implicitness score** | | | |
| | **0.69** | **0.84** | **1.06** | **1.37** |

The equation for computing Kendall's Tau when measuring the correlation of two ranks:

$$\tau = \frac{(\text{Number of concordant pairs}) - (\text{Number of discordant pairs})}{(\text{number of pairs})} \tag{11}$$

## A.4 HYPERPARAMETER SENSITIVITY

Please see the results in Fig. 10.

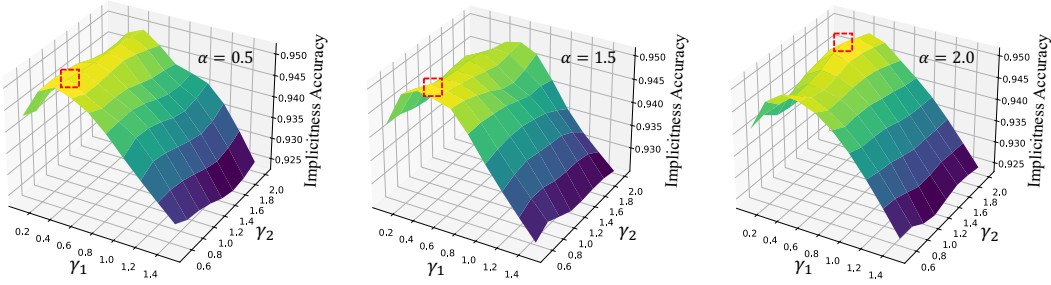

Figure 10: Hyperparameter sensitivity of $\gamma_1$ and $\gamma_2$ on Implicitness Accuracy when $\alpha = \{0.5, 1.5, 2.0\}$. The highest point is marked in red dotted box.

## A.5 USER STUDY DESIGN

**OOD Data Topics** We adopted most of the topics from the book *Crucial conversations: tools for talking when stakes are high* (Kohnen, 2008), section "*Some Common Crucial Conversations*" in *Chapter One*. This book introduces how to apply different expressions in important conversations, which we think is a very useful resource. Other topics were designed by ourselves. The details are shown in Tab. 11. We also asked a linguist to assess the quality of sentences under each group in Tab. 11. The quality of the test examples was evaluated on two dimensions: pragmatics and implicitness. The pragmatics rating ($0$ to $4$) reflects the level of pragmatic agreement among the sentences. If all four sentences convey the same meaning, the group receives a score of $4$. If three of the sentences share the same meaning, the group is given a score of $3$, and so on. In terms of the implicitness rating, since $\binom{4}{2} = 6$ pairs can be created from the four sentences in a group, the implicitness rating will range from $0$ to $6$, measuring the extent to which the four sentences show a progressively higher level of implicitness.

**Platforms and Participant Recruitment** We used Gorilla (https://app.gorilla.sc/) to design the interface for the user study and Prolific (https://www.prolific.com/) for recruiting participants. All participants were native and proficient speakers of English, hold a bachelor's degree, have conducted more than $10$ surveys on Prolific, and have a history of $100\%$ approval rate.

**User Study Quality Assurance** We provided participants with examples of both the *Ranking* and *Choice* tasks before they began the survey, showcasing the tasks and guiding them on how to properly answer the questions. Their progress was monitored at every step to ensure no abnormal responses were submitted. Before recruiting participants, we asked two lab members, who were unfamiliar with the project, to pilot the study. While the two pilot participants did not report any questions or concerns when completing the experiment, they noted that the judgments were highly subjective.

Table 11: The source of each OOD topic group is either borrowed from or inspired by the book *Crucial conversations: tools for talking when stakes are high* or developed by us. Detailed sentences under each group, after refinement and verification, are shown in Tab.8. For each group, a linguist provided a assessment of the pragmatics quality (rated from 0 to 4) and implicitness level quality (rated from 0 to 6) of its sentences.

| Topic Group | Source | Pragmatics Rating | Implicitness Rating |
|---|---|---|---|
| $G_1$: Ending Relationship | Book | 4 | 6 |
| $G_2$: Critiquing a colleague's work | Book | 4 | 6 |
| $G_3$: Dealing with a rebellious child | Book | 4 | 6 |
| $G_4$: Giving the boss feedback about her behavior | Book | 4 | 6 |
| $G_5$: Giving an unfavorable performance review of ballet | Book | 4 | 6 |
| $G_6$: Asking a roommate to move out | Book | 4 | 6 |
| $G_7$: Handing off a difficult project to a colleague | Original | 4 | 6 |
| $G_8$: Disliking John's personality | Book | 4 | 6 |
| $G_9$: Decline a friend's party invitation | Original | 4 | 6 |
| $G_{10}$: Remind a roommate to clean the kitchen | Book | 4 | 6 |

## A.6 EMBEDDING VISUALIZATION AND CASE STUDY ON OOD DATA

Fig. 11 displays a 2-D visualization of the pragmatic embeddings for sentences from the ten topic groups in user study data using t-SNE. The clusters of sentences within the same topic group are evident, highlighting the effectiveness of the pragmatic embeddings in grouping similar thematic content. Fig. 12 presents the distribution of their semantic embeddings. Although points marked with the same colors generally tend to cluster together, some are noticeably distant from their corresponding groups. We analyze this is attributed to the fact that while sentences within the same topic may be pragmatically aligned, their varied expressive styles contribute to a broader semantic distribution. For better understanding, a case study of the semantic embeddings from two topic groups is provided in Fig. 13.

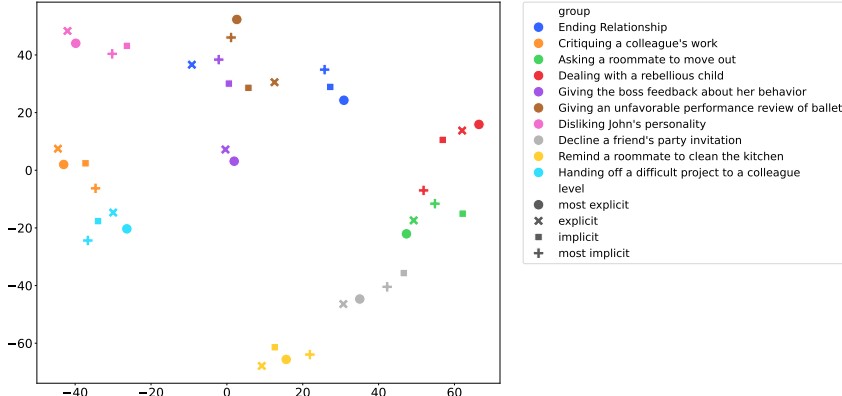

Figure 11: t-SNE Visualization of pragmatic Embeddings of the OOD Data in the User Study. Points with the same color belong to the same topic group, and different markers indicate varying levels of implicitness.

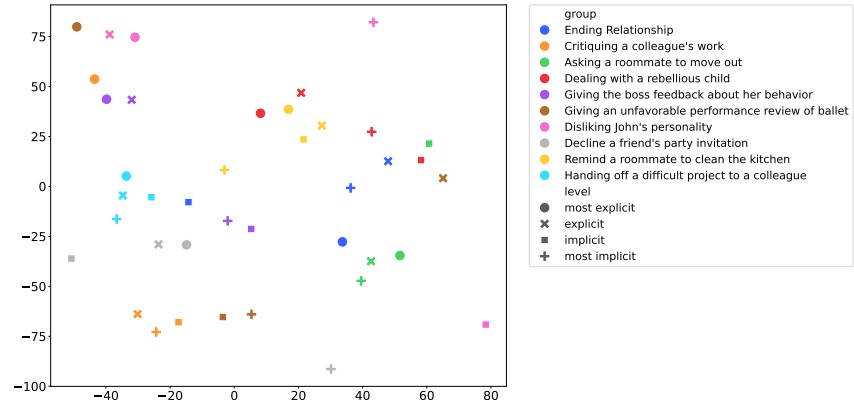

Figure 12: t-SNE Visualization of semantic embeddings of the OOD data in the User Study.

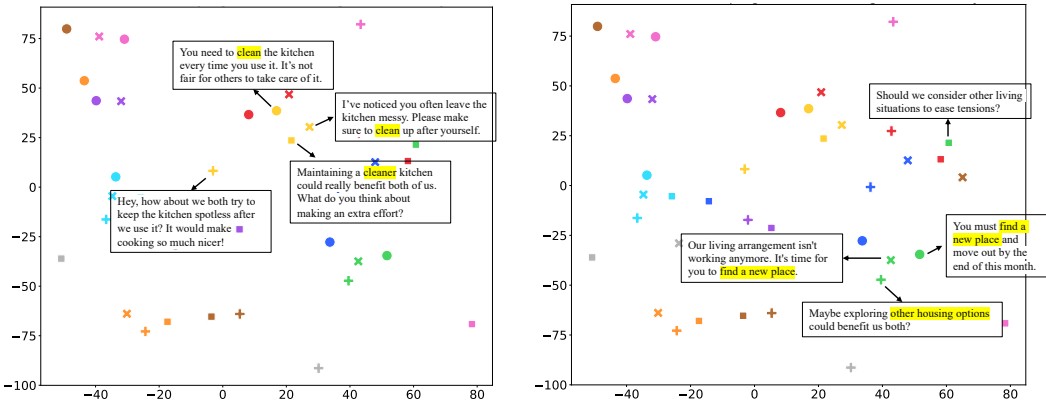

Figure 13: Case studies on the distribution of semantic embeddings for two topic groups: *Remind a roommate to clean the kitchen* (left figure) and *Asking a roommate to move out* (right figure). The corresponding sentence for each point is annotated alongside. The text pieces highlighted within the sentences are the parts conveying similar semantics, explaining why some sentences cluster together while others do not.

## A.7 IMPLICITNESS SCORE DISTRIBUTION OF HATE SPEECH DATASETS

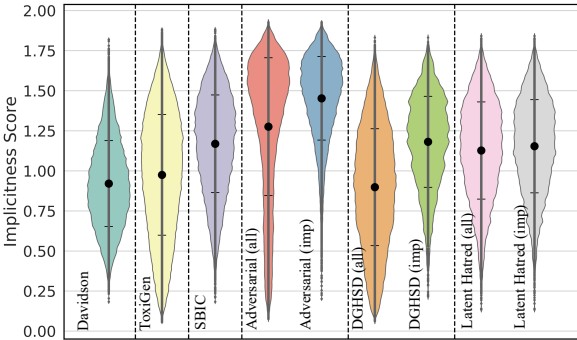

Figure 14: Distribution of implicitness scores computed by IMPSCORE of each hate speech detection dataset.

## A.8 PROMPTS FOR HATE SPEECH DETECTION

> **role**: system
> **content**: Please indicate whether the following text is toxic, harmful, or otherwise inappropriate by simply answering "yes" or "no."
>
> **role**: user
> **content**: <Inserted Hate Speech Sentence>

Figure 15: Prompts for using GPT-4-Turbo and Llama-3.1-8B-Instruct for Hate Speech Detection. We set the maximum output token of Llama-3.1-8B-Instruct to 3. All other hyperparameters are set to default.

## A.9 SAMPLES AND HATE SPEECH DETECTION ACCURACY ON ADVERSARIAL DATASET

The hate speech detection result on Adversarial dataset is shown in Fig. 16. The implicitness range $[1.75, 2]$ of Adversarial dataset is filled with very similar sentences shown in Tab. 12.

Table 12: Examples of very similar sentences in the implicitness range $[1.75, 2]$ in Adversarial dataset.

| |
|---|
| middle east is a war zone , why do we clearly have to admit a failing cultural into ours ? |
| middle east is a war zone , why do we apparently have to admit a failing cultural into ours ? |
| middle east is a war zone , why do we surely have to admit a failing cultural into ours ? |
| middle east is a war zone , why do we doubtlessly have to admit a failing cultural into ours ? |
| middle east is a war zone , why do we presumably have to admit a failing cultural into ours ? |
| middle east is a war zone , why do we indeed have to admit a failing cultural into ours ? |
| middle east is a war zone , why do we probably have to admit a failing cultural into ours ? |
| a woman should undoubtedly be aloud to wear what she wants , it is a little thing in the western world we like to call freedom . |
| a woman should doubtlessly be aloud to wear what she wants , it is a little thing in the western world we like to call freedom . |
| a woman should seemingly be aloud to wear what she wants , it is a little thing in the western world we like to call freedom . |
| a woman should likely be aloud to wear what she wants , it is a little thing in the western world we like to call freedom . |
| a woman should clearly be aloud to wear what she wants , it is a little thing in the western world we like to call freedom . |
| a woman should absolutely be aloud to wear what she wants , it is a little thing in the western world we like to call freedom . |

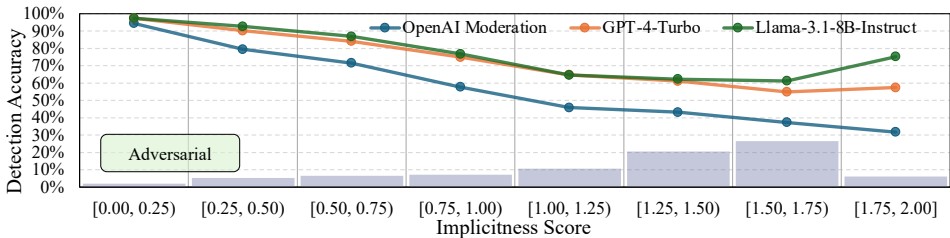

Figure 16: Hate speech detection accuracy of LLMs on sentence samples with different levels of implicitness. Blue columns indicate the distribution of samples among different ranges of implicitness.

## A.10 USER STUDY QUESTIONNAIRE

Fig. 17 shows a screenshot of a question in the *Ranking* task, and Fig. 18 shows a screenshot of a question in the *Choice* task.

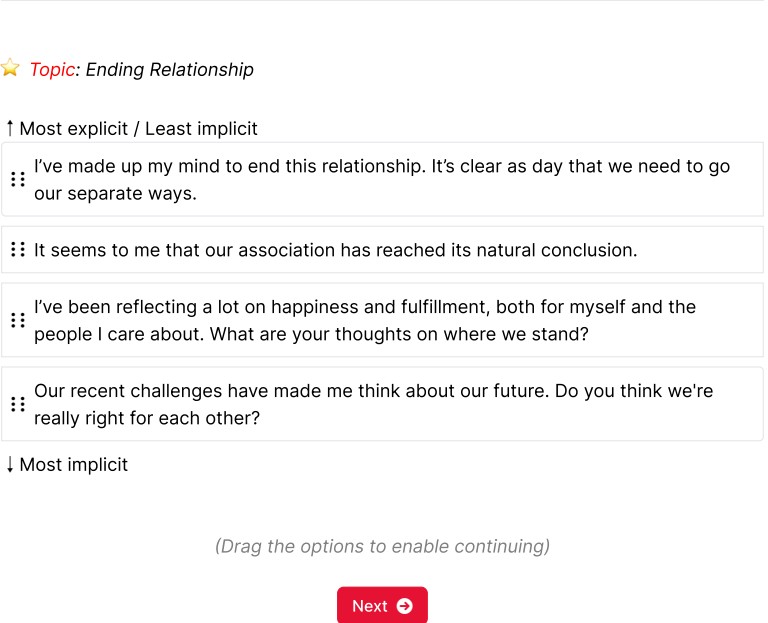

Figure 17: Screenshot of a *Ranking* task question in user study questionnaire.

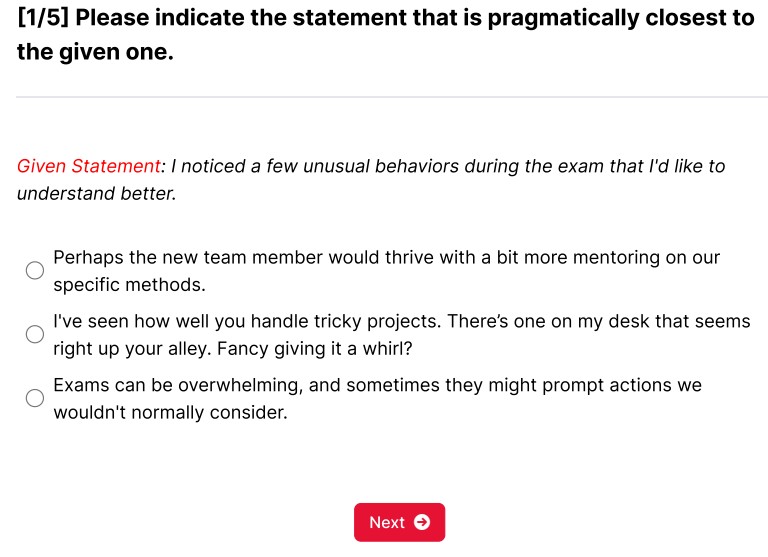

Figure 18: Screenshot of a *Choice* task question in user study questionnaire.

## A.11 TRAINING PERFORMANCES BREAKDOWN

Table 13 includes the performance breakdown of IMPSCORE in different data sources in the test set during training.

Table 13: Breakdown of IMPSCORE's performance on different data sources in the test set during training.

| Source of Test Data | Source Topics | # (implicit, explicit) Pair | # Correct Prediction | Accuracy |
|---|---|---|---|---|
| DGHSID | Implicit Hate Speech Detection | 302 | 245 | 0.811 |
| Latent Hatred | Implicit Hate Speech Detection | 1166 | 1156 | 0.991 |
| ToxiGEN | Implicit Hate Speech Detection | 270 | 224 | 0.830 |
| Adversarial | Implicit Hate Speech Detection | 2518 | 2460 | 0.977 |
| SBIC | Implicit Hate Speech Detection | 900 | 896 | 0.996 |
| Implicatures | Natural Language Inference | 89 | 68 | 0.764 |
| ImpPres | Natural Language Inference | 12 | 12 | 1.000 |
| LUDWIG | Natural Language Inference | 69 | 55 | 0.797 |
| ABSA | Sentiment Analysis | 136 | 112 | 0.824 |
| SemEval | Irony / Sarcasm Detection | 132 | 108 | 0.818 |
| Snarks | Irony / Sarcasm Detection | 16 | 10 | 0.625 |
| R-3 | Irony / Sarcasm Detection | 14 | 12 | 0.857 |
| PDTB | Irony / Sarcasm Detection | 6 | 4 | 0.667 |

## A.12 CROSS-LINGUAL PERFORMANCE OF IMPSCORE

We conducted an experiment to evaluate IMPSCORE 's performance on three non-English languages: French, German, and Chinese. Specifically, we first utilized Google Translate[6] to translate the 10 group topics from the user study in §6 into these non-English languages. Subsequently, we prompted ChatGPT in each non-English language to generate four sentences per topic, varying in levels of implicitness. Here are the prompt formats:

1. French: Pour ce sujet « Mettre fin à une relation », pourriez-vous générer 4 phrases avec différents niveaux d'implicit ? du plus explicite au plus implicite.
2. German: Formulieren Sie zum Thema „Beziehung beenden" bitte 4 Sätze mit unterschiedlichen Impliziheitsgraden, vom explizitesten bis zum implizitesten.
3. Chinese: 对于这个主题"结束关系。", 你能生成 4 个具有不同隐含程度的句子吗？ （从隐含程度最低到最高）

The generated sentences in these languages were then translated back to English using Google Translate. We used Google Translate instead of directly using ChatGPT for translation to minimize inherent cross-language alignment biases in ChatGPT, ensuring more independence of each language. Then, we use IMPSCORE to rank the four sentences within each topic and compare the predictions with the gold rankings. The average results are summarized below. Detailed scores for each group are visible in the Tab. 14.

These findings indicate that IMPSCORE demonstrates a moderate positive correlation with gold rankings in these translated sentences. However, its accuracy diminishes compared to the assessment on original English sentences in our paper (avg. $\tau = 0.76$ and $\rho = 0.84$), particularly for language that is linguistically distant from English (*i.e.*, Chinese).

This reduction in accuracy may stem from a loss of subtle implicitness nuances in original languages during translation into English, suggesting potential limitations in using translations as a proxy for cross-lingual evaluation of implicitness.

Table 14: IMPSCORE's performance on translated text of non-English sentences.

| Non-English Languages | Avg. $\tau$ over 10 groups | Avg. $\rho$ over 10 groups |
|---|---|---|
| French | 0.63 | 0.74 |
| German | 0.63 | 0.70 |
| Chinese | 0.43 | 0.58 |

---

[6]https://translate.google.com/

## A.13 ADDITIONAL USER STUDY RESULTS

We conducted another 10 groups OOD test data for user study, following the same procedure detailed in the paper in §6. Below are the topics:

- $G_1$: Reject an auto dealerships offer
- $G_2$: Talking to a coworker about a specific hygiene problem
- $G_3$: A student cheating on the exam
- $G_4$: Asking a friend to repay a loan
- $G_5$: Complain about micromanaging
- $G_6$: Unpleasant food smell
- $G_7$: Criticizing a colleague's work
- $G_8$: Eating with big sound
- $G_9$: Having bad memory about hometown
- $G_{10}$: Talking to a team member who isn't keep commitments

We divided the topics into two sets (Set 1: topics $G_1$–$G_5$; Set 2: topics $G_6$–$G_{10}$) and recruited five human participants to rank their implicitness. The results of their predictions compared to the gold ranks are summarized in Tab. 15. The correlation plots between human participants and IMPSCORE are in Fig. 19 and Fig. 20.

The results indicate a generally high accuracy of IMPSCORE with the gold ranks and concordant with human participants, which align with our conclusion in line 406-409 in the paper. In set 1, IMPSCORE slightly underperforms the average human ranking, and in set 2, IMPSCORE slightly surpasses the average human ranking.

Table 15: Performances of IMPSCORE and human participants compared to the gold ranks on additional 10 data groups.

| Set | Human 1 | | Human 2 | | Human 3 | | Human 4 | | Human 5 | | IMPSCORE | |
|---|---|---|---|---|---|---|---|---|---|---|---|---|
| | Avg. $\tau^\uparrow$ | Avg. $\rho^\uparrow$ | Avg. $\tau^\uparrow$ | Avg. $\rho^\uparrow$ | Avg. $\tau^\uparrow$ | Avg. $\rho^\uparrow$ | Avg. $\tau^\uparrow$ | Avg. $\rho^\uparrow$ | Avg. $\tau^\uparrow$ | Avg. $\rho^\uparrow$ | Avg. $\tau^\uparrow$ | Avg. $\rho^\uparrow$ |
| 1 | 0.87 | 0.92 | 1.00 | 1.00 | 0.93 | 0.96 | 0.87 | 0.92 | 0.87 | 0.92 | 0.80 | 0.88 |
| 2 | 0.87 | 0.92 | 0.93 | 0.96 | 0.87 | 0.92 | 0.67 | 0.72 | 0.93 | 0.96 | 0.93 | 0.96 |

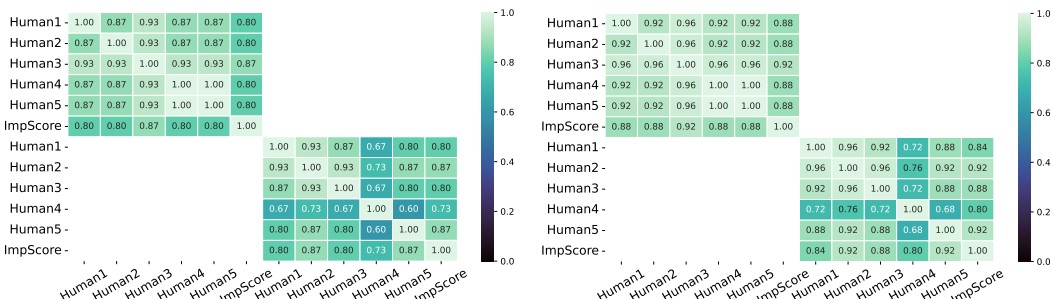

Figure 19: $\tau$ correlation values between participants.

Figure 20: $\rho$ correlation values between participants.

## A.14 QUALITATIVE ANALYSIS OF IMPSCORE'S PERFORMANCE

We evaluated the IMPSCORE's performance on the in-distribution test set on the following features.

### A.14.1 THE LENGTH OF THE SENTENCES

For $5,630$ pairs of (*implicit sentence*, *explicit sentence*) in the test data, we analyzed sentence length by calculating the average number of space-separated tokens per sentence. We then plotted the

distribution of sentence lengths alongside the corresponding numbers of correctly predicted cases. The result plot is in Fig. 21, which shows no significant decline in accuracy for longer sentences.

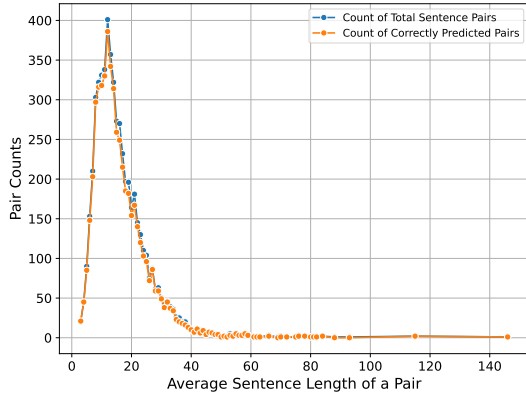

Figure 21: IMPSCORE's prediction performance on different length of sentence pairs.

### A.14.2 SENTIMENT

We evaluated the sentiment of each sentence in the pairs using the pre-trained sentiment classification model `cardiffnlp/twitter-roberta-base-sentiment-latest`. Sentiments were categorized into {positive, neutral, negative}, and each pair was grouped into one of eight combinations (*e.g.*, negative-positive, neutral-neutral).

The result plot is in Tab. 22. The "negative-positive" group exhibits the lowest performance, likely due to a bias introduced by its small sample size. Otherwise, the results do not reveal a clear underperformance for any specific sentiment group. Additionally, we observed that negative sentiment dominates the dataset—a trend that aligns with our expectations, as implicit sentences often arise in sensitive or negative contexts.

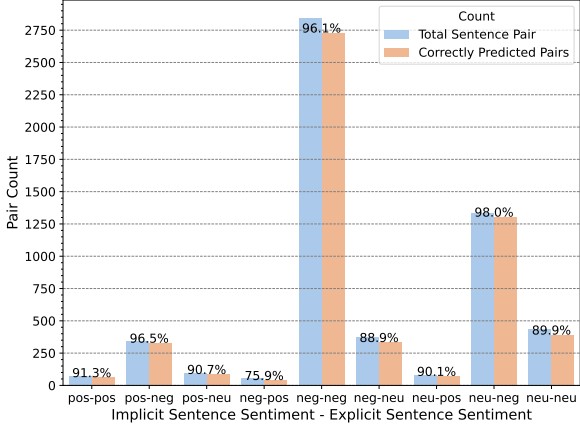

Figure 22: IMPSCORE's prediction performance on different sentiment of sentence pairs.

### A.14.3 TSNE FOR CORRECT/INCORRECT PREDICTIONS

We also analyzed the semantic distribution of correctly predicted pairs and incorrectly predicted pairs. To do so, we concatenated the implicit and explicit sentences in each pair, generated embeddings using Sentence-BERT, and applied t-SNE for dimensionality reduction to visualize the results in a 2D plot. The visualization is in Fig. 23.

The result reveals that while both correct and incorrect predictions are broadly distributed, certain clusters of incorrect predictions emerge (highlighted as Box 1 and Box 2 in the plot). On closer

examination, we found that the incorrect predictions in Box 1 predominantly pertain to digital devices, such as "computers," "CDs," "laptops," and "RAM." This suggests a potential challenge for IMPSCORE in handling domain-specific or technical terminology. Box 2 contains pairs where distinguishing the implicitness levels between sentences proved exceptionally difficult. For example: (- *Will that make everything alright? - Thank you.*, - *Will that make everything alright? - Thank you for your concern.*)

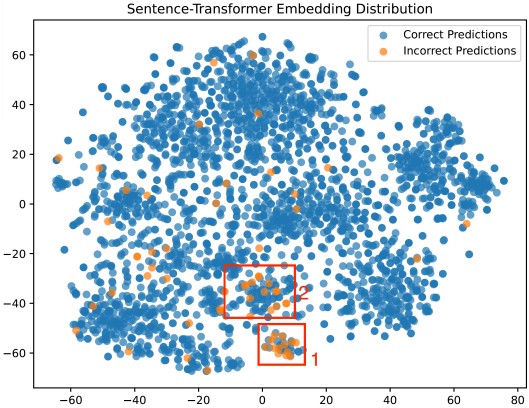

Figure 23: Embedding distributions of IMPSCORE's correctly and incorrectly predicted sentence pairs.

### A.15    APPLYING IMPSCORE TO THE GENERATED SENTENCES OF LLMS

We zero-shot prompted seven LLMs (GPT-3.5, GPT-4o, Llama-3.2-Instruct, Mistral-7B, Mistral-Nemo, Claude-haiku, and Claude-sonnet) with 10 topics same with the ones in the user study and asked them to generate four sentences with different implicitness levels. We used the prompt "*Given the topic [topic], please generate four sentences about it, ranging from most explicit to most implicit. Output each sentence on a new line.*" and set the temperature to each model's default value.

For each run, we calculate the average implicitness scores of sentences over 10 groups in each implicitness level (most explicit, explicit, implicit, most implicit). We conducted 5 runs and plot the mean and standard deviation of the 5 average scores. The result is in Tab. 24.

The results demonstrate that, according to IMPSCORE's evaluation, all the models are generally capable of producing sentences with distinguishable implicitness levels. Among them, GPT-4o exhibited the best performance in generating sentences with the most clearly distinguishable levels of implicitness. Additionally, GPT-4o and Claude-sonnet showed the strongest ability to generate highly implicit sentences, aligning with our expectations given their advanced capabilities.

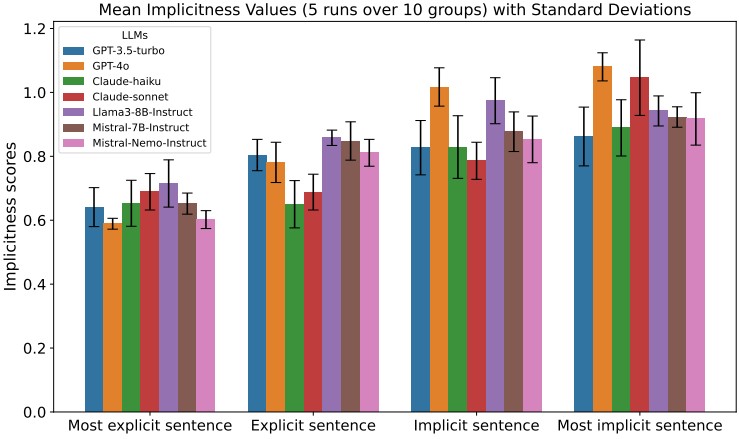

Figure 24: IMPSCORE's scores on the generated sentences of seven LLMs.

