# OpenReview forum: "ImpScore: A Learnable Metric For Quantifying The Implicitness Level of Sentences"
_ICLR.cc/2025/Conference — ICLR 2025 Spotlight_

### Official Review · Reviewer_bQJm · 2024-11-02

**Soundness:** 3
**Presentation:** 4
**Contribution:** 3
**Rating:** 8
**Confidence:** 4

**Summary:**

This paper introduces a scalar metric to compute the implicitness score for sentences known as IMPSCORE. They also contribute a dataset of 56,290 pairs of implicit and explicit sentences used to train the IMPSCORE model. The metrics reliability is studied through a user study on OOD data. Moreover, the authors apply IMPSCORE on various hate speech benchmarks and also compare the performance of three language models - gpt-4-turbo, llama-3.1-8b-instruct and OpenAI Moderation - on these datasets with varying levels of implicitness.

**Strengths:**

1. The paper tries to address an important problem of understanding the implicitness level of language through a novel framework to measure it based on the divergence between semantic meaning and pragmatic interpretation of a sentence. It also introduces a novel dataset consisting of diverse samples (validated from a linguistic expert) to train the model used in computing the implicitness score.
2. Overall, the study is comprehensive. Authors justify their choice for the final setting by performing sufficient ablations in terms of the design choices and hyper-parameters. They also conduct a user study to understand correlation of the metric with human judgement.
3. The paper is technically sound and is well written in general.

**Weaknesses:**

1. The number of sentences used for human judgment correlation (40 - 4 sentences across 10 topics) are not sufficient enough to examine the generalization ability of the IMPSCORE metric. Expanding this set is necessary to improve the reliability of the metric.
2. Even though some justification is provided to measure the pragmatic distance (in the Introduction section), it is not entirely clear on why it is important. The ablation experiments also does not seem to affect the pragmatic distances, further casting doubts on its significance. Additional explanation for this sub-metric would be beneficial.

**Questions:**

1. Why is the same hyper-parameter $\gamma_1$ used for both $L_{imp} (s_1,s_2)$ and $L_{imp} (s_1,s_3)$?
2. Is the data split into train, validation and test sets stratified i.e. does it maintain the ratio of positive and negative pairs?

---

> ### Author Response · Authors · 2024-11-21
> **Thank you for your review. Here is our response**
>
> Thank you for recognizing the contributions of our work and for providing valuable feedback. We have made a response to each of your comments and questions. We hope our responses clarify your concerns and enhance your impression of our paper.
>
> **To Weakness 1: Limited User Study**
>
> We acknowledge this limitation regarding the sample size in our user study and we also discussed it in the final section of our paper. When designing user study, our primary focus was to assess the correlation between ImpScore’s predictions and human judgments, which led us to prioritize sufficient evaluations per group over a larger number of groups.
>
> However, we agree that expanding the study to include a larger and more diverse sample would further strengthen the validation of ImpScore. Thus, we conducted another 10 groups OOD test data for user study, following the same procedure detailed in the paper. Below are the topics and corresponding results:
>
> 10 Topics:
> - Reject an auto dealerships offer.
> - Talking to a coworker about a specific hygiene problem.
> - A student cheating on the exam
> - Asking a friend to repay a loan
> - Complain about micromanaging
> - Unpleasant food smell
> - Criticizing a colleague's work
> - Eating with big sound
> - Having bad memory about hometown
> - Talking to a team member who isn't keep commitments
>
> The specific four sentences for each topic are available **in this [link](https://anonymous.4open.science/r/iclr25_rebuttal-35A1/Additional%20User%20Study/additional_user_study_groups.csv)**.
>
> We divided the topics into two sets (Set 1: topics 1–5; Set 2: topics 6–10) and recruited five human participants to rank their implicitness. The results of their predictions compared to the gold ranks are summarized in the below table. The correlation plots between participants and ImpScore are **in these links: [$\tau$ tau plot](https://anonymous.4open.science/r/iclr25_rebuttal-35A1/Additional%20User%20Study/ood_heatmap_additional_tau.pdf) and [$\rho$ rho plot](https://anonymous.4open.science/r/iclr25_rebuttal-35A1/Additional%20User%20Study/ood_heatmap_additional_rho.pdf)**.
>
> | Set | Human1 ($\tau$) | Human1 ($\rho$) | Human2 ($\tau$) | Human2 ($\rho$) | Human3 ($\tau$) | Human3 ($\rho$) | Human4 ($\tau$) | Human4 ($\rho$) | Human5 ($\tau$) | Human5 ($\rho$) | ImpScore ($\tau$) | ImpScore ($\rho$) |
> |-------|-------|-------|-------|-------|-------|-------|-------|-------|-------|-------|-------|-------|
> | 1  | 0.87 | 0.92 | 1.00 | 1.00 | 0.93 | 0.96 | 0.87 | 0.92 | 0.87 | 0.92 | 0.80 | 0.88 |
> | 2  | 0.87 | 0.92 | 0.93 | 0.96 | 0.87 | 0.92 | 0.67 | 0.72 | 0.93 | 0.96 | 0.93 | 0.96 |
>
> The results indicate a generally high accuracy of ImpScore with the gold ranks and concordant with human participants, which align with our conclusion in line 406-409 in the paper. In set1, ImpScore slightly underperforms the average human ranking, and in set 2, ImpScore slightly surpasses the average human ranking.
>
> We have included this result in Appendix A.13 in the revised version of our paper, highlighted in blue.
>
> Nonetheless, we recognize that larger-scale, more comprehensive evaluations could provide further insights and robustness, and we plan to include such studies in future work.
>
> **To Weakness 2: The Design of Pragmatic Distance**
>
> Thanks for raising this question. We believe that comparing the implicitness levels of sentences with similar meanings is crucial for a nuanced understanding of their relative implicitness level, and the pragmatic distance metric serves to interpret these comparisons effectively.
>
> The lack of notable differences in training performances with changes to $\gamma_2$ and $\alpha$ may be attributed to the use of Euclidean distance for calculating pragmatic distance. This approach enables the model to generate a wide range of pragmatic distances to optimize the loss function effectively. We have added a detailed explanation on this in the revised version of our paper.
>
> Moreover, we believe the design of the pragmatic loss is of value, as it offers benefits and enables a range of applications. For example, it allows for testing the consistency of text pieces within a group expressing similar ideas.

---

> ### Author Response · Authors · 2024-11-21
> **Response From Authors (2)**
>
> **To Q1: Same Hyper-parameters**
>
> The hyperparameter $\gamma_1$ represents the minimum gap that the metric needs to learn for distinguishing between implicit and explicit sentences, regardless of whether they appear in positive or negative pairs. It’s challenging to precisely predetermine the gap for each sentence pair by setting training labels and hyperparameters, and the only thing we know is the relative implicitness between two sentences. The model itself learns the precise gap size through training, so we set $\gamma_1$ consistently for both positive and negative pairs to guide this learning process.
>
> **To Q2: Data Split**
>
> Yes, the ratios remain the same in the train, validation, and test sets. The number of positive and negative pairs is the same in the training set because each input unit consists of a positive pair and a corresponding negative pair. During data splitting, we split by units, so the validation and test sets still contain the same number of positive and negative pairs. We examine the implicitness level for both types of pairs during validation and testing.

---

> > ### Comment · Reviewer_bQJm · 2024-11-25
> > **Thank you for your responses!**
> >
> > Thank you for addressing my concerns about the user study and reinforcing the importance of pragmatic distance. Your responses are satisfactory and all the best for your paper.

---

> > > ### Author Response · Authors · 2024-11-25
> > > **Thank you!**
> > >
> > > Thank you again for your insightful feedback and valuable contribution to ICLR!
> > >
> > > Sincerely,
> > >
> > > Authors

---

### Official Review · Reviewer_keRt · 2024-11-02

**Soundness:** 2
**Presentation:** 3
**Contribution:** 3
**Rating:** 6
**Confidence:** 4

**Summary:**

This paper introduces ImpScore, a reference-free metric designed to quantify the level of implicitness in human language. The model for ImpScore is trained via contrastive learning by calculating the divergence between the semantic (literal) meaning and the pragmatic (contextual) interpretation of a sentence. They collected a training data with 112,580 pairs. Through extensive experiments, ImpScore demonstrates strong correlation with human judgments. They also show the utility in evaluating hate speech detection datasets, revealing limitations in current large language models' ability to handle implicit content.

**Strengths:**

1. The paper tackles a rarely addressed problem, quantifying implicitness in language. It offers a new approach to measuring implicitness that moves beyond binary or purely lexical classifications.
2. ImpScore is a reference-free metric, which distinguishes it from previous metrics that often rely on external references or manual annotations.

**Weaknesses:**

1. Limited scope of implicit languages. The paper’s dataset includes synthetic data generated by GPT-3.5 and focuses primarily on specific types of implicit language (e.g., hate speech). However, ImpScore’s generalizability across broader types of implicit expressions (such as indirect requests or cultural idioms) remains uninvestigated. I wonder if the authors have evaluated their model on other types of implicit expressions, such as irony or casual dialogues.
2. While the paper performs some ablation studies, it primarily uses Sentence-BERT as the embedding model. Exploring alternative text encoders could provide a more comprehensive understanding of ImpScore’s design choices. They may also utilize some contrastive learning-based models, such as SimCSE.
3. I am also concerned about their data synthetic approach. The reliance on GPT-3.5 for generating explicit counterparts of implicit sentences could introduce unintended biases, particularly if the model generates stereotypical or simplified responses (we can also see the average length of explicit sentences is much shorter than implicit ones). I wonder if the authors verified the quality of the generated explicit sentences.
4. I also expect some model comparisons. For example, they can use their training data as a binary classification task and compare the classification accuracy between the simple classifier and their proposed model.

**Questions:**

* How does the performance of ImpScore vary across different types of implicit language (e.g., indirect requests, cultural idioms, irony)? Are there specific types that it struggles with?
* Have you examined any potential biases and quality of the synthetic data?
* Can you compare with existing methods or other baselines for implicitness? For example, the methods mentioned in Related Work (e.g. Garassino et al. (2022) and Vallauri & Masia (2014))?
* I would like to see if you can implement ImpScore to the LLMs' abilities to generate implicit expressions. Similar to the data creation in Section 6.1, you can prompt different LLMs to generate 4 sentences with varying implicitness levels. Then, you calculate the ImpScore for these generations.

---

> ### Author Response · Authors · 2024-11-21
> **Thank you for your review. Here is our response**
>
> We sincerely thank the reviewer for their detailed feedback and thoughtful critique of our work. We appreciate your recognition of the strengths of our paper, including the novel approach to quantifying implicitness in language and the reference-free nature of ImpScore. Below, we provide responses to each of the points raised.
>
> **Response to Weakness 1: Limited Scope of Implicit Languages**
>
> Thank you for highlighting this concern. We made considerable efforts to collect diverse and representative data for training ImpScore, resulting in a dataset spanning 13 different sources and covering five types of implicit language, including irony (see Table 6 in the paper). However, we acknowledge that the availability of well-labeled resources for implicit language remains limited, which poses a significant challenge. Among the domains where such data exist, hate speech detection is particularly well-studied, leading to its prominence in our dataset. This reflects the data landscape rather than a deliberate bias in our approach.
>
> We agree that expanding the training data to encompass a broader range of implicit expressions is an important future direction. One promising approach involves leveraging LLMs to generate synthetic data while ensuring diversity and coherence. This direction is emphasized in the revised paper and highlighted in blue in the "Conclusion and Future Work" section of the revised paper.
>
> Additionally, we conducted a breakdown analysis of ImpScore’s performance across various types of implicit language within the in-distribution test set. Please see our response to Q1 for detailed results.
>
> **Response to Q1: Performance Across Different Types of Implicit Language**
>
> We agree that a comprehensive analysis of ImpScore’s performance across different types of implicit language would be insightful. In the table below, we provide a breakdown of ImpScore’s performance on our in-distribution test data, which encompasses a diverse range of topics. Please refer to the table below.
>
>
> | Source of Test Data | Source Topics                 | # (implicit, explicit) Test Sentence Pair | # Correct Prediction | Accuracy |
> |----------------------|-------------------------------|-------------------------------------------|-----------------------|----------|
> | DGHSID              | Implicit Hate Speech Detection| 302                                       | 245                   | 0.811    |
> | Latent Hatred       | Implicit Hate Speech Detection| 1166                                      | 1156                  | 0.991    |
> | ToxiGEN             | Implicit Hate Speech Detection| 270                                       | 224                   | 0.830    |
> | Adversarial         | Implicit Hate Speech Detection| 2518                                      | 2460                  | 0.977    |
> | SBIC                | Implicit Hate Speech Detection| 900                                       | 896                   | 0.996    |
> | Implicatures        | Natural Language Inference    | 89                                        | 68                    | 0.764    |
> | ImpPres             | Natural Language Inference    | 12                                        | 12                    | 1.000    |
> | LUDWIG              | Natural Language Inference    | 69                                        | 55                    | 0.797    |
> | ABSA                | Sentiment Analysis            | 136                                       | 112                   | 0.824    |
> | SemEval             | Irony / Sarcasm Detection     | 132                                       | 108                   | 0.818    |
> | Snarks              | Irony / Sarcasm Detection     | 16                                        | 10                    | 0.625    |
> | R-3                 | Irony / Sarcasm Detection     | 14                                        | 12                    | 0.857    |
> | PDTB                | Discourse Relation            | 6                                         | 4                     | 0.667    |
>
> The results show consistently high accuracy (>0.8) across most topics. However, underperformance in certain datasets (e.g., Snarks and PDTB) can be attributed to their small training and test set sizes, a limitation we have discussed in the paper.
>
> Furthermore, our out-of-distribution user study includes a broad set of topics like "Dealing with a rebellious child" and "Ending a relationship," which differ substantially from the training set, particularly the Hate Speech data. ImpScore performed well on these topics, demonstrating strong alignment with human judgments.
>
> Moving forward, we plan to enhance ImpScore by training it with larger and more diverse data sources.

---

> ### Author Response · Authors · 2024-11-21
> **Response From Authors (2)**
>
> **To Weakness 2: Exploring alternative text encoders**
>
> At the outset, we selected Sentence-BERT as the text encoder because it intuitively seemed well-suited to our task. This choice was validated by its strong performance, achieving over 95% accuracy during training. Given its strong performance, we were not strongly motivated to explore alternative encoders to further improve accuracy. Instead, we focused on another aspect of our model — the ability to independently model semantics and pragmatics, which is the core innovation of ImpScore. Additionally, as binary text classification is a relatively simple task, we believe the remaining ~5% inaccuracy is more likely attributable to limitations or noise in the dataset rather than the encoder itself. Sentence-BERT also offers the practical advantage of being lightweight and efficient.
>
> That said, we recognize the value of investigating different text encoders, such as SimCSE, especially as the training set grows in size and complexity. To address this, we have added a discussion in the "Conclusion and Future Work" section of our revised paper.
>
> **To Q2 and Weakness 3: Potential biases and quality of the training set**
>
> Thanks for asking this question. **We've already asked a linguist for a qualitative review in the paper**, which we have detailed in Appendix A.2 (specifically on Page 17). While we used GPT to generate explicit sentences for a portion of the training data, not all data was GPT-generated, as shown in Table 6 in the Appendix. The reason we use GPT is that we found some datasets containing only labeled implicit sentences, so we had to generate corresponding explicit sentences for training.
>
> We are mindful of potential biases introduced by the training data. To mitigate this, we sourced data from diverse domains (17 datasets across 5 topics). However, we found limited text datasets with well-defined labels for implicit language, with Hate Speech datasets forming a substantial portion of the training data. We are committed to expanding this diversity by including more data from additional domains in future training to enhance ImpScore’s scope.
>
> **To Weakness 4 and Q3: Model comparison**
>
> Thank you for raising this concern. As for comparing with relevant baselines, to the best of our knowledge, no directly comparable models for calculating implicitness scores or specifically trained for classifying sentences by implicitness level exist in the past work. We carefully reviewed the two works Garassino et al. (2022) and Vallauri & Masia (2014) which fall under the field of pragmatics. It is impossible to directly use them as baseline models for comparison. In particular,
> - Garassino et al. (2022): It qualitatively assessed participants' abilities to detect and interpret different types of implicit content. However, it does not provide a quantitative measure or scale for the level of implicitness in the sentences used.
> - Vallauri & Masia (2014): It studied implicit causality bias, which is more focused on how specific verbs suggest causal relationships, making it different from our task. Their study calculates scores based on how often people link causality to the subject or object of a verb (you can read their results [here](https://www.sciencedirect.com/science/article/pii/S0378216613002282#tbl0095)), while we focus on scoring the implicitness of entire sentences.
>
> We have revised the discussion of these works in the Related Work section of our paper to clarify their scope and distinguish them from our contributions.
>
> That said, we acknowledge that models from related fields, such as textual entailment, could be considered for indirect comparison. In the textual entailment task, models predict whether one sentence can infer another, a concept we discuss in Section 4. We considered that entailment models might indirectly assess implicitness, as an implicit sentence often implies its explicit counterpart. To test this, we use a powerful pre-trained textual entailment model roberta-large-mnli https://huggingface.co/FacebookAI/roberta-large-mnli to predict implicitness in sentence pairs by determining whether the implicit sentence could entail the explicit one. However, this approach achieved an accuracy of only 44.94%, highlighting the uniqueness of our task and the limitations of applying related methods to it directly.
>
> Additionally, we compared ImpScore’s results with human judgments, which we believe serve as a meaningful baseline for evaluating ImpScore’s performance.

---

> ### Author Response · Authors · 2024-11-21
> **Response From Authors (3)**
>
> **To Q4: Apply ImpScore to the generated text of LLMs**
>
> This is a good idea to test the application of ImpScore. Following your recommendation, we zero-shot prompted 7 LLMs (GPT-3.5, GPT-4o, Llama-3.2-Instruct, Mistral-7B, Mistral-Nemo, Claude-haiku, and Claude-sonnet) with 10 topics same with the ones in the user study and ask them to generate 4 sentences with different implicitness levels. We used the following prompt format and set the temperature to each model’s default value:
>
> > "Given the topic
> <topic>, please generate four sentences about it, ranging from most explicit to most implicit. Output each sentence on a new line."
>
> For each run, we calculate the average implicitness scores of sentences over 10 groups in each implicitness level (most explicit, explicit, implicit, most implicit). We conducted 5 runs and plot the mean and standard deviation of the 5 average scores. **The result is available in this [link](https://anonymous.4open.science/r/iclr25_rebuttal-35A1/generative%20application/llms_generation.pdf)**.
>
> The results demonstrate that, according to ImpScore’s evaluation, all the models are generally capable of producing sentences with distinguishable implicitness levels. Among them, GPT-4o exhibited the best performance in generating sentences with the most clearly distinguishable levels of implicitness. Additionally, GPT-4o and Claude-sonnet showed the strongest ability to generate highly implicit sentences, aligning with our expectations given their advanced capabilities.
>
> We have included this analysis in Appendix A.15 in the revised version of our paper, highlighted in blue.
>
> ---
>
> These are our responses to your comments and questions. We hope they address your concerns and improve your view of our paper. We hope you might consider improving your rating, as it would mean a lot to us.

---

> ### Comment · Reviewer_keRt · 2024-11-23
> **follow up question.**
>
> Thanks for your reply.
>
> Could you remind me how you converted your ImpScore into the labels of these classification tasks in your response to Q1?

---

> ### Author Response · Authors · 2024-11-23
> **Reply to your follow up question**
>
> We appreciate your engagement with our responses and for providing additional feedback.
>
> During metric training, the test set comprises pairs of (*implicit sentence*, *explicit sentence*). ImpScore calculates an implicitness score within the range [0, 2] to each sentence and compares the scores of the implicit and explicit sentences in each pair. If the score for the implicit sentence is higher than that of the explicit sentence, ImpScore successfully identifies the higher level of implicitness, indicating a correct classification. This is how ImpScore works during test. We hope this helps you recall our experiments.

---

> > ### Comment · Reviewer_keRt · 2024-11-24
> > **Thanks for your response.**
> >
> > I found that my concerns have been addressed. I have adjusted my rate accordingly.

---

> > > ### Author Response · Authors · 2024-11-24
> > > **Thank you!**
> > >
> > > We are glad to have addressed your concerns. Thank you once again for your valuable feedback, which has helped improve our work, and for your contribution to ICLR!
> > >
> > > Sincerely,
> > >
> > > Authors

---

### Official Review · Reviewer_czyP · 2024-11-03

**Soundness:** 4
**Presentation:** 4
**Contribution:** 3
**Rating:** 8
**Confidence:** 4

**Summary:**

This paper introduces IMPSCORE, a novel metric for quantifying the implicitness level of language by measuring the divergence between semantic meaning and pragmatic interpretation. The authors develop an interpretable regression model trained through pairwise contrastive learning on a newly created large dataset of 112,580 sentence pairs. The metric processes sentences using a text encoder to generate embeddings, maps semantic and pragmatic features into separate spaces, and calculates implicitness scores based on the divergence between these features after space transformation.

Experimental results validate the reliability of IMPSCORE. The authors also conduct a user study comparing the metric's assessments with human evaluations on out-of-distribution data, showing strong correlation with human judgments. They also demonstrate practical applications by analyzing implicitness levels in hate speech detection datasets and evaluating performance of several large language models across different implicitness levels, revealing that model performance degrades as implicitness increases.

**Strengths:**

- The paper is well motivated and presents a clear operational definition of implicitness and develops a concrete methodology for measuring it. The paper is well written and easy to follow.
- Experimental results show strong support for the IMPSCORE as a reliable metric. The authors further validate it through careful ablation studies, user studies, and practical applications in analyzing both datasets and model performance.
- The training dataset is substantial and diverse (and is a contribution in itself), drawing from multiple domains and types of implicit language, while the evaluation includes both in-distribution and out-of-distribution testing.

**Weaknesses:**

- The metric is developed and tested only for English language content. It is not clear how this would translate to other languages or whether this metric will work in cross-lingual settings.
- While well-designed, the user study involves only 10 participants evaluating 10 questions each, which is a relatively small sample size for validating a metric intended for broad use.

**Questions:**

- I'm curious if you could run some experiments to test the metric's performance on other languages? An easy (though, not perfect) way would be to look at translated sentence pairs across multiple languages to assess its language independence and potential biases?
- I'd love to see some qualitative error analysis for the metric. Are there any patterns that emerge in the sentences on which IMPSCORE isn't accurate?
- Is IMPSCORE just as reliable on longer sentences as on shorter ones?

---

> ### Author Response · Authors · 2024-11-21
> **Thank you for your review. Here is our response**
>
> We sincerely thank the reviewer for their thoughtful and detailed feedback, as well as their recognition of the strengths of our work, including the clarity of our operational definition of implicitness, the strong experimental validation of ImpScore, and the substantial contributions of our curated dataset and user study. Your insights have been invaluable in refining and strengthening our submission.
>
> **Response to Weakness 1 and Question 1: Cross-lingual Setting**
>
> We agree with the reviewer that extending ImpScore to multilingual settings is an important direction for future work. This effort would involve constructing multilingual training datasets and developing comprehensive test cases to evaluate performance across different languages. Additionally, the text encoder must support the target language, though this is increasingly straightforward given that most modern text encoders are inherently multilingual. We have included a discussion of these considerations in the revised manuscript, highlighted in blue in the final section.
>
> Additionally, to provide preliminary insights into cross-lingual applicability, we conducted an experiment evaluating ImpScore’s performance on three non-English languages: **French**, **German**, and **Chinese**.
>
> * Experimental Setup: We translated the 10 topics from our user study into these languages using Google Translate. Next, ChatGPT was prompted in the respective language to generate four sentences per topic, each with varying levels of implicitness (see prompts below). The sentences were then back-translated into English to minimize alignment biases.
>
> Prompts:
> - **French**: Pour ce sujet « Mettre fin à une relation », pourriez-vous générer 4 phrases avec différents niveaux d'implicit? du plus explicite au plus implicite.
> - **German**: Formulieren Sie zum Thema „Beziehung beenden“ bitte 4 Sätze mit unterschiedlichen Implizitheitsgraden, vom explizitesten bis zum implizitesten.
> - **Chinese**: 对于这个主题“结束关系。”，你能生成 4 个具有不同隐含程度的句子吗？（从隐含程度最低到最高）
>
> We used Google Translate instead of directly using ChatGPT for translation to minimize inherent cross-language alignment biases in ChatGPT, ensuring more independence of each language. All the generated sentences are available **in this [folder](https://anonymous.4open.science/r/iclr25_rebuttal-35A1/non-English%20test%20results/)** for your review.
>
>
> * Results: ImpScore demonstrated a moderate positive correlation with gold rankings for French and German (average τ = 0.63, ρ = 0.74 and 0.70, respectively) but showed reduced performance for Chinese (τ = 0.43, ρ = 0.58) (see table below for full results).  These findings indicate that ImpScore achieves a moderate positive correlation with gold rankings for the translated sentences. However, its accuracy declines compared to the results on the original English sentences reported in our paper (avg. $\tau$ = 0.76 and $\rho$ = 0.84), particularly for linguistically distant languages such as Chinese. This reduction in accuracy likely stems from the loss of subtle implicitness nuances during translation into English, highlighting potential limitations of using translations as a proxy for cross-lingual evaluation of implicitness.
>
> | non-English Language | Avg. $\tau$ over 10 groups  | Avg. $\rho$ over 10 groups  |
> |------------|------------|------------|
> | French | 0.63 | 0.74 |
> | German | 0.63 | 0.70 |
> | Chinese | 0.43 | 0.58 |
>
>
> This experiment and its findings are included in Appendix A.12, highlighted in blue.

---

> ### Author Response · Authors · 2024-11-21
> **Response From Authors (2)**
>
> **Response to Weakness 2: User Study**
>
> We acknowledge the reviewer’s observation regarding the relatively small sample size of our user study. Our initial design prioritized robust evaluations within each group, but we agree that a larger and more diverse participant base could further validate ImpScore.
> To address this, we conducted an additional user study with 10 new out-of-distribution (OOD) topics, as shown below. These topics were split into two sets (Set 1: topics 1–5; Set 2: topics 6–10), with five human participants ranking the implicitness of four sentences per topic.
>
> 10 Topics:
> - Reject an auto dealerships offer.
> - Talking to a coworker about a specific hygiene problem.
> - A student cheating on the exam
> - Asking a friend to repay a loan
> - Complain about micromanaging
> - Unpleasant food smell
> - Criticizing a colleague's work
> - Eating with big sound
> - Having bad memory about hometown
> - Talking to a team member who isn't keep commitments
>
> The specific four sentences for each topic are available for your review **here [link](https://anonymous.4open.science/r/iclr25_rebuttal-35A1/Additional%20User%20Study/additional_user_study_groups.csv)**.
>
> * Results: The rankings provided by participants and ImpScore strongly correlated with the gold rankings. In Set 1, ImpScore slightly underperformed the average human ranking (τ = 0.80, ρ = 0.88), whereas in Set 2, it slightly surpassed human performance (τ = 0.93, ρ = 0.96). These results demonstrate that ImpScore generally achieves high accuracy relative to the gold rankings and aligns well with human judgments, consistent with our conclusions in lines 406–409 of the paper.
> The full results are summarized in the table below. Additionally, correlation plots between participants and ImpScore can be found **at the following links: [$\tau$ tau plot](https://anonymous.4open.science/r/iclr25_rebuttal-35A1/Additional%20User%20Study/ood_heatmap_additional_tau.pdf) and [$\rho$ rho plot](https://anonymous.4open.science/r/iclr25_rebuttal-35A1/Additional%20User%20Study/ood_heatmap_additional_rho.pdf)**.
>
> | Set | Human1 ($\tau$) | Human1 ($\rho$) | Human2 ($\tau$) | Human2 ($\rho$) | Human3 ($\tau$) | Human3 ($\rho$) | Human4 ($\tau$) | Human4 ($\rho$) | Human5 ($\tau$) | Human5 ($\rho$) | ImpScore ($\tau$) | ImpScore ($\rho$) |
> |-------|-------|-------|-------|-------|-------|-------|-------|-------|-------|-------|-------|-------|
> | 1  | 0.87 | 0.92 | 1.00 | 1.00 | 0.93 | 0.96 | 0.87 | 0.92 | 0.87 | 0.92 | 0.80 | 0.88 |
> | 2  | 0.87 | 0.92 | 0.93 | 0.96 | 0.87 | 0.92 | 0.67 | 0.72 | 0.93 | 0.96 | 0.93 | 0.96 |
>
> This additional study further supports the robustness of ImpScore, and the details are included in Appendix A.13, highlighted in blue. Nonetheless, we plan to conduct larger-scale evaluations in future work.

---

> ### Author Response · Authors · 2024-11-21
> **Response From Authors (3)**
>
> **Response to Q2 and Q3: Qualitative Error Analysis for the Metric**
>
> We appreciate the reviewer’s suggestion for a qualitative analysis to deepen our understanding of ImpScore’s performance. Below are the analyses we conducted:
>
> **Sentence Length**:
> We analyzed sentence length for 5,630 pairs of (implicit sentence, explicit sentence) in the test data by calculating the average number of space-separated tokens per sentence. The distribution of sentence lengths and the corresponding number of correctly predicted cases were plotted.
> The results, **available here, [link](https://anonymous.4open.science/r/iclr25_rebuttal-35A1/features/sentence_length.pdf)**, show no significant decline in accuracy for longer sentences. **This analysis also addresses the concern raised in Q3**.
>
> **Sentiment**:
> Sentences were categorized into sentiment combinations (e.g., positive-positive, negative-neutral). While the “negative-positive” group exhibited slightly lower performance due to the small sample size, overall results showed no notable sentiment-related biases. A detailed plot is provided in the appendix.
>
> Using the pre-trained sentiment classification model cardiffnlp/twitter-roberta-base-sentiment-latest, we evaluated the sentiment of each sentence. Sentiments were categorized into {positive, neutral, negative}, and the pairs were grouped into eight combinations (e.g., negative-positive, neutral-neutral).
>
> The results, **visualized here [link](https://anonymous.4open.science/r/iclr25_rebuttal-35A1/features/sentiment.pdf)**, reveal that the "negative-positive" group exhibited the lowest performance, likely due to its small sample size. Otherwise, no specific sentiment group showed a clear decline in performance. Additionally, we observed that negative sentiment dominates the dataset, which aligns with expectations since implicit sentences often appear in sensitive or negative/hostile contexts.
>
>
> **t-SNE Analysis**:
> We also analyzed the semantic distribution of correctly and incorrectly predicted pairs by concatenating the implicit and explicit sentences in each pair, generating embeddings with Sentence-Transformer, and applying t-SNE for dimensionality reduction. The 2D visualization can **be accessed here [link](https://anonymous.4open.science/r/iclr25_rebuttal-35A1/features/tnse_plot.pdf)**.
>
> The visualization shows that while correct and incorrect predictions are broadly distributed, certain clusters of errors emerge. Specifically:
>
> Box 1: Contains errors related to domain-specific or technical terminology, such as "computers," "CDs," "laptops," and "RAM," suggesting challenges for ImpScore in handling such cases.
>
> Box 2: Includes pairs where distinguishing implicitness levels was exceptionally difficult. For instance:
> "Will that make everything alright? - Thank you."
> "Will that make everything alright? - Thank you for your concern."
>
> These findings highlight areas for potential improvement in future iterations of ImpScore.
>
> All these analyses and the corresponding visualizations are included in Appendix A.14, highlighted in blue.
>
> ------
>
> We hope these responses and revisions address the reviewer’s concerns. We are grateful for your constructive feedback, which has significantly enhanced our work.

---

> > ### Comment · Reviewer_czyP · 2024-11-23
> > **This is great, thank you!**
> >
> > You've addressed all the points I raised and honestly I have nothing else to say other than that I would love to see this paper at the conference.

---

> ### Author Response · Authors · 2024-11-23
> **Thank you!**
>
> Thank you again for your insightful feedback and valuable contribution to ICLR!
>
> Sincerely,
>
> Authors

---

### Author Response · Authors · 2024-11-26
**Response to all reviewers**

We just uploaded a new revision of our paper where we corrected a number of format errors in table and figure captions.

-----

We sincerely thank all reviewers for their thoughtful and detailed feedback on our work. Reviewers highlighted the significance of the problem addressed in this paper—quantifying the level of implicitness in language—describing it as "an important problem" (bQJm) and "rarely addressed" (keRt) in existing literature. They also commended our novel approach to measuring implicitness as "operationally clear and concrete" (czyP). Moreover, two reviewers (czyP, bQJm) commended the robustness and comprehensiveness of our methodology, particularly noting the sufficiency of our experiments, which include ablation studies, user studies, and practical applications.

The constructed training dataset was similarly praised for its significant and diverse nature, with reviewers describing it as “a novel dataset consisting of diverse samples” (bQJm) and “substantial and diverse” (czyP), underscoring its value as an additional contribution to the field. Additionally, reviewer (keRt) highlighted the reference-free nature of ImpScore, emphasizing how it "distinguishes it from previous metrics."

-----

We have replied to reviewers' every question and concern and conducted additional experiments to incorporate their useful feedback. Key efforts include:
1. *More User Study (reviewer czyP and bQJm)*: We followed reviewers' suggestions and conducted an additional user study with 10 new out-of-distribution (OOD) topics. The results are consistent with our conclusions in the original manuscript.
2. *Breakdown on ImpScore’s performance (reviewer czyP and keRt)*: we conducted further analysis on ImpScore’s performance on different implicitness types and sentences with different features such as length and sentiments.
3. *Additional Application of ImpScore (reviewer keRt)*: We applied ImpScore to the generated text of 7 LLMs (GPT-3.5, GPT-4o, Llama-3.2-Instruct, Mistral-7B, Mistral-Nemo, Claude-haiku, and Claude-sonnet) and presented the comparison on their implicitness levels.

We have expanded our Appendix and extended our discussion in the Conclusion and Future Work section to incorporate these changes and reviewers' feedback.

Sincerely,

Authors

---

### Meta-Review · Area_Chair_Gj7Z · 2024-12-17

**Metareview:**

This work introduces ImpScore, a new metric for measuring the implicitness of language in text. Specifically, ImpScore measures the difference between the surface semantic meaning and the expected pragmatic interpretation of a given text by training a regression model on a set of (embeddings of) sentence pairs. To train this model, the paper also proposes a large dataset of these sentence pairs that map between implicit and explicit phrasings of a sentence's meaning.

The experimental results show that the ImpScore model is very accurate on the proposed dataset (i.e., good at distinguishing between implicit and explicit sentences) and correlates well with human judgments. The experiments also consider practical settings for measuring the implicitness of language with hate-speech detection by LLMs (where models perfrom worse at detecting hate-speech at high levels of implicitness).

Strengths:
- Proposes a clear, operational definition of implicitness and provides a large dataset with examples of implicit and explicit sentence pairs following this definition (czyPs, bQJm).
- The paper is well-written and well-motivated (czyP, bQJm).
- The experimental results indicate that this is a reliable metric for measuring implicitness, based on agreement with human judgments and ablations. The paper also suggests how this score can be used in practical settings by ablating the effect of implicitness on hate-speech detection (czyPs).

Weaknesses:
There are no major weaknesses, in my opinion. Minor weaknesses include:
- The use of synthetic data during dataset creation (keRt), though the paper and author response indicate that significant effort was taken to verify the automatically generated data that went into the new dataset.
- It is unclear if the dataset and model will be released with the paper. I recommend that the authors release these artifacts, which will greatly strengthen the contribution of this work.

Other weaknesses, such as the limited scope of the metric (czyPs, keRt) and small sample size for user study (zyPs, bQJm), were addressed in the author response. I appreciate the translation-based multilingual evaluation here and would love to see follow-up work in this area with annotated data in other languages.

**Additional Comments On Reviewer Discussion:**

The authors provided detailed responses to each reviewer. The reviewer with the most concerns raised their score after the response period; the other reviewers already recommended acceptance.

---

### Decision · Program_Chairs · 2025-01-22

Accept (Spotlight)